# Zishen Yutai Pill increased live births in advanced maternal age women: a randomized clinical trial

Yu Li[1,2], Fei Gong[3], Xiaohong Wang[4], Linli Hu[5], Hong Li[6], Yichun Guan [7], Hong Ye[8], Li Fan[9], Haiyan Bai[10], Ze Wang[11], Wei Huang[12], Xiaoling Ma[13], Dongzi Yang [1] ✉ & Heping Zhang [14] ✉

Advanced maternal age (AMA, ≥35 years) women undergoing assisted reproductive technology (ART) face reduced live birth rates (LBR) and remain a major clinical challenge. In a large randomized trial, Zishen Yutai Pill (ZYP) improved LBR in the general population, and a subsequent post hoc analysis suggested efficacy in AMA women, though it was underpowered to draw firm conclusions. To address the issue, we conducted a multicenter, prospective, double-blind, placebo-controlled, randomized trial (NCT03703700) to evaluate whether ZYP increases LBR in AMA women. Women aged 35-42 years with BMI < 28 kg/m² at 12 tertiary-level hospitals in China were randomly assigned to receive ZYP or placebo orally (5 g once, three times daily) from day 19-23 of the preceding menstrual cycle until 2 weeks after embryo transfer, continuing to 5 weeks post transfer if biochemical pregnancy was confirmed. The primary outcome was fresh-cycle LBR. A total of 1467 participants (734 ZYP, 733 placebo) were enrolled. In the intention-to-treat analysis, live birth occurred in 23.3% (171/734) with ZYP vs. 19.0% (139/733) with placebo (relative ratio 1.23 [95% CI, 1.01-1.50]; absolute difference 4.3% [95% CI, 0.2-8.5]; *P* = 0.042). This study showed that Zishen Yutai Pill increased live birth rates in women aged 35–42 undergoing IVF, without elevating the risk of maternal or neonatal adverse events. (NCT03703700, https://clinicaltrials.gov/study/NCT03703700)

Advanced maternal age (AMA) is defined as the maternal age of 35 years or older at the time of delivery[1]. AMA women are more inclined to use assisted reproductive technology (ART) due to the age-related infertility[2]. In recent years, the proportion of AMA women undergoing autologous ART increased from 63.0% in 2014 to 71.8% in 2019 around the globe[3,4]. Nevertheless, advanced age is conclusively associated with a reduced live birth rate (LBR) during ART. According to the 2019 national survey by the United States Centers for Disease Control and Prevention, the LBR of ART declined from approximately 42%-44% for women under 34 years old to 40.5% at age 35, 31.0% at 40, and dropped below 13.0% for those aged 45 and older[5]. AMA women exhibit several

age-related reproductive disadvantages, including diminished ovarian reserve as reflected by lower follicle counts and oocyte quality, a higher prevalence of uterine structural abnormalities (e.g., fibroids, adenomyosis), reduced endometrial receptivity, and altered hormonal profiles (such as decreased progesterone levels and receptor expression), compared to non-AMA women[6–8].

In China, the pregnancy needs of AMA women have been sparked by the implementation of "two-child" and "three-child" policies in recent years[9,10]. Over the past decade, the proportion of AMA mothers has nearly doubled in mainland China, from 9.9% in 2012 to 18.0% in 2022[11,12]. Recent guidelines in China recommend specific practices for

AMA women undergoing ART, including interventions aimed at improving ovarian reserve, such as the use of growth hormone (GH) and dehydroepiandrosterone (DHEA)[13]. Additional interventions like lifestyle management, antioxidant supplements, and acupuncture were also reported to exert positive effects on ovarian reserve and endometrial receptivity[14]. However, while many of these interventions demonstrated potential to improve ovarian responses, endometrial indices, hormonal levels, and aging-related biomarkers in AMA women, few studies specifically evaluated their effects on live birth in this population[7,15–18].

Zishen Yutai Pill (ZYP) is a representative traditional Chinese medicine (TCM) preparation applied in female reproductive health[19]. ZYP was recently reported to improve the fresh cycle LBR among women undergoing in vitro fertilization/intracytoplasmic sperm injection (IVF/ICSI)[20]. Post hoc analyses indicated that ZYP intervention resulted in significant improvements in pregnancy outcomes among the AMA population, both in fresh embryo transfer (ET) cycles and freeze-thawed ET cycles[21,22]. Preclinical studies suggested that ZYP enhanced endometrial receptivity by upregulating key implantation markers such as HOXA10, ICAM1, and SPP1[23,24]. Clinical evidence in patients with diminished ovarian reserve indicated its potential to improve embryo quality via modulation of follicular-fluid BMP15 and GDF9 expression[25]. The post hoc subgroup analyses from our previous trial provided promising evidence for the effect of ZYP on AMA women in ART (ZYP vs. placebo: clinical pregnancy rate, 33.0% [68/206] vs. 23.1% [51/221], absolute difference and 95% confidence interval [CI]: 9.9% [1.4% ~ 18.3%], $P = 0.022$; LBR, 26.2% [54/206] vs. 19.9% [44/221], absolute difference and 95% CI: 6.3% [−1.7% ~ 14.3%], $P = 0.122$)[21]. Nevertheless, conclusive data were still needed to confirm the effectiveness of ZYP in ART settings among the AMA population due to the increasing population of AMA women in current ART practice and their poor pregnancy outcome prognosis. Thus, the present study aimed to assess the effectiveness and safety of ZYP in enhancing pregnancy outcomes among AMA women undergoing IVF/ICSI procedures. Specifically, this multicenter, prospective, double-blind, placebo-controlled, randomized trial compared ZYP versus placebo in infertile women aged 35–42 years undergoing IVF/ICSI at 12 tertiary hospitals in China, to determine whether ZYP increased LBR in fresh ET cycles, and to evaluate its effects on ovarian stimulation (OS) outcomes, biochemical pregnancy rate, implantation rate, clinical pregnancy rate, miscarriage rate, and maternal, fetal, and neonatal safety.

## Results

### Participants
Recruitment was conducted from March 4, 2019 to October 14, 2023. Follow-up was completed in September 14, 2024. We screened 1673 women after pre-screening hospital records. A total of 1467 women (87.7%) were included in the intention-to-treat (ITT) analysis, who were randomly assigned to receive ZYP (734 women) or placebo (733 women) (Fig. 1). The baseline characteristics of the participants were similar between the two arms (Table 1 and Supplementary Table 7), as were the characteristics of embryo transfer procedures and OS outcomes (Table 2 and Supplementary Table 8). A total of 633 women (43.1%) did not receive embryo transfer or dropped out of the study (312 in the ZYP group and 321 in the placebo group) before ET. The main reasons for not undergoing embryo transfer are presented in Fig. 1. The overall proportion of cycle cancellation and drop-out before ET was similar between groups ($P = 0.619$). Additionally, the live birth outcome of 1461 participants (99.6%) were retrieved. Six participants (3 participants in the ZYP group, 3 participants in the placebo group) were lost to follow-up after ET, and were included in the ITT analysis but treated as no live birth delivery (Fig. 1).

### Clinical efficacy
In the primary analysis, the LBR was 23.3% (171 of 734) in the ZYP group, and 19.0% (139 of 733) in the placebo group (RR, 1.23; 95% CI, 1.01 ~ 1.50; $P = 0.042$; Table 3). Results of the per-protocol (PP) analyses were similar with the ITT analysis (41.0% vs. 33.7%, RR 1.22; 95% CI, 1.01 ~ 1.47; $P = 0.040$; Table 4).

In the ITT analysis (Table 3), compared with the placebo group, the ZYP group showed increases in the biochemical pregnancy rate (31.6% vs. 26.6%; RR, 1.19; 95% CI, 1.01 ~ 1.40; $P = 0.035$), implantation rate (36.0% vs. 30.6%; RR, 1.18; 95% CI, 1.01 ~ 1.37; $P = 0.034$) and clinical pregnancy rate (29.4% vs. 24.1%; RR, 1.22; 95% CI, 1.03 ~ 1.45; $P = 0.022$).

Results of the PP analyses were similar with the ITT analyses (Table 4). The intervention of ZYP resulted in significant elevation in all secondary pregnancy outcomes, including rates of biochemical pregnancy (56.1% vs. 48.1%; RR 1.17; 95% CI, 1.02 ~ 1.34; $P = 0.029$), implantation (36.8% vs. 30.7%; RR 1.20; 95% CI, 1.02 ~ 1.40; $P = 0.023$), clinical pregnancy (51.9% vs. 43.3%; RR 1.20; 95% CI, 1.03 ~ 1.39; $P = 0.019$).

In both ITT and PP analyses, there were no significant differences in the rates of pregnancy loss (all $P > 0.05$) (Tables 3 and 4). No significant differences were observed with regard to the birth weight and gestation age.

Among women aged 35–37, the intervention of ZYP resulted in significant differences in the rates of live birth (28.9% vs. 21.1%; RR 1.37; 95% CI, 1.08 ~ 1.74; $P = 0.009$), biochemical pregnancy (35.8% vs. 29.3%; RR 1.22; 95% CI, 1.00 ~ 1.48; $P = 0.046$), implantation (41.3% vs. 32.8%; RR 1.26; 95% CI, 1.05 ~ 1.51; $P = 0.013$), clinical pregnancy (34.1% vs. 26.1%; RR 1.31; 95% CI, 1.06 ~ 1.61; $P = 0.010$) (Supplementary Table 1). No significant differences were observed with regard to the rates of pregnancy loss among women aged 35–37 (all $P > 0.05$). No significant differences were observed with regard to rates of live birth, biochemical pregnancy, implantation, clinical pregnancy and pregnancy loss among women aged 38-39 or 40–42 (all $P > 0.05$) (Supplementary Table 1).

### Safety
No significant differences were observed with regard to the rates of moderate or severe ovarian hyperstimulation syndrome (OHSS), ectopic pregnancy, pregnancy complications (all $P > 0.05$). No significant differences were observed in fetal and neonatal adverse events (all $P > 0.05$) (Supplementary Table 5).

## Discussion
In this RCT involving Chinese patients undergoing IVF/ICSI-ET, ZYP as an adjunctive therapy around the time of embryo transfer significantly increased the fresh cycle LBR. Additionally, ZYP showed beneficial effects on secondary pregnancy outcomes, including the biochemical pregnancy, implantation and clinical pregnancy rates. No differences were observed in OS outcomes, the rate of pregnancy loss and other maternal, fetal and neonatal adverse events.

The ZYP was approved for the treatment of threatened miscarriage and recurrent pregnancy loss[26,27]. In our previous RCT, ZYP demonstrated a trend toward increased LBR among AMA women during their fresh IVF/ICSI-ET cycles in the post hoc subgroup analysis[21]. This finding may not be reliable and fail to accurately represent the true effect size of ZYP. Furthermore, women were all treated with various exogenous hormones during their fresh ET cycles. A subgroup analysis with a limited sample size could not fully reflect the effects of ZYP on AMA women throughout the entire ART process.

In the current research, we validated the effectiveness of ZYP on fresh cycle live births among AMA women. Our analyses showed robust results in the ITT population (Table 3), PP population (Table 4), and embryo transfer population (Supplementary Table 2). As supplemental evidence, we also pooled and analyzed data from both our current and previous trials. The pooled analysis results (Supplementary Table 3 and 4) demonstrated consistent effect sizes, further

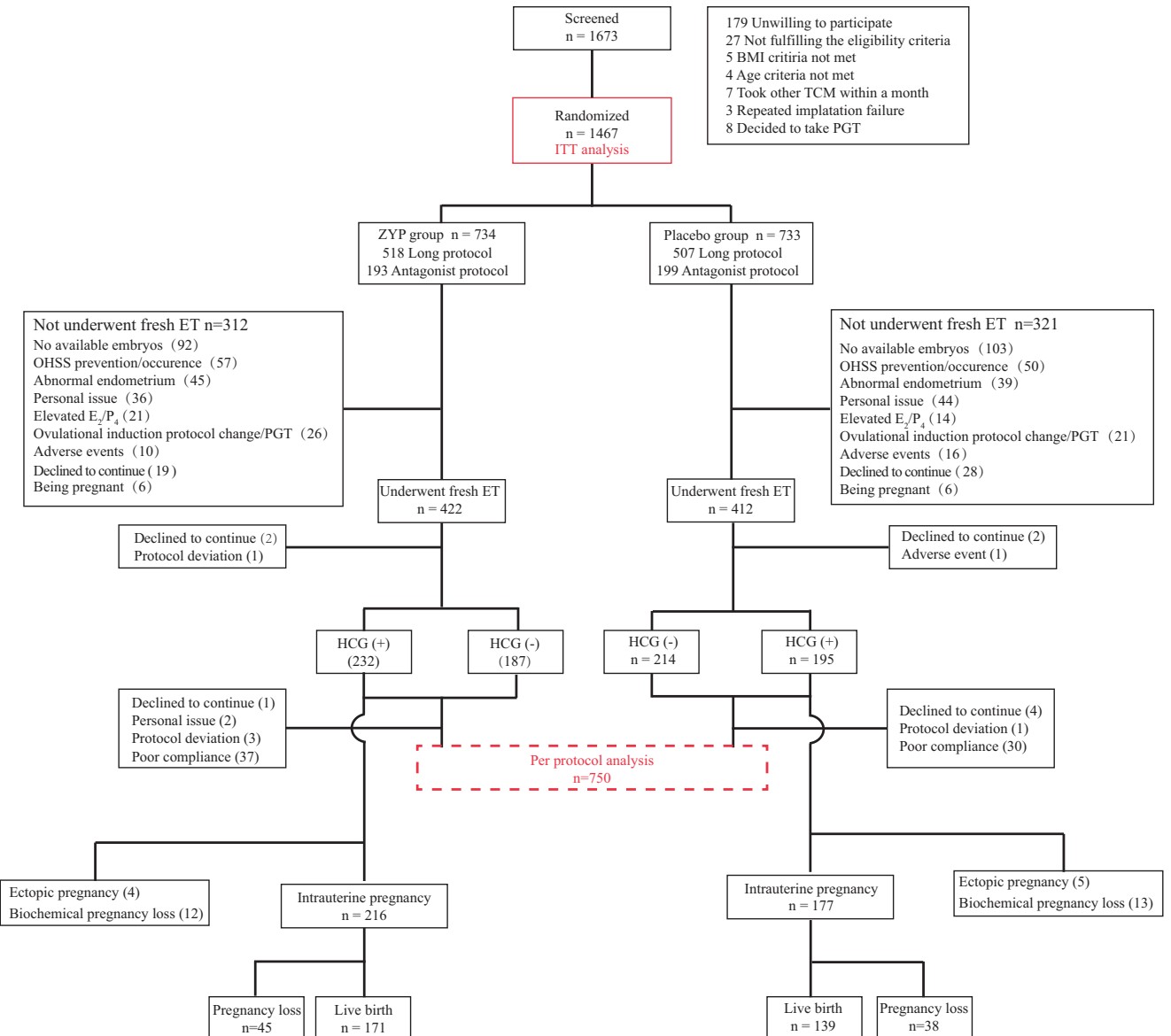

**Fig. 1 | Study flowchart.** The inner pathway illustrates participant progression according to the IVF cycle timeline. The peripheral boxes summarize exclusions applied in the ITT and PP analyses. *ZYP* Zishen Yutai Pill; *ITT* intention-to-treat; *PP* per-protocol; *BMI* body mass index; *TCM* traditional Chinese medicine; PGT, pre-implantation genetic testing; *ET* embryo transfer; *HCG* human chorionic gonadotropin; *OHSS* ovarian hyperstimulation syndrome; $E_2$ estradiol; $P_4$ progesterone. Note: Six pregnancies occurred after randomization but before ET in each arm ($n = 12$ in total). These pregnancies were unrelated to the ET procedures and,

therefore excluded from the denominator when calculating fresh cycle live birth and other pregnancy-related outcomes. "Abnormal endometrium" was defined as the presence of any of the following conditions prior to embryo transfer: thin endometrium ($< 7$ mm), uterine cavity effusion, or abnormal uterine bleeding. "Elevated $E_2/P_4$" was defined as serum $E_2$ levels $\geq 3000$ pg/ml and/or serum $P_4$ levels $\geq 1.5$ ng/ml. HCG (+) represents a biochemical pregnancy test outcome as positive, whereas HCG (-) as negative.

supporting the findings. Additional post hoc analyses adjusted for site and age further supported our findings (Supplementary Tables 9–11).

Aging poses significant challenges to pregnancy outcomes in ART by affecting various facets such as ovarian reserve, oocyte/embryo quality, and endometrial receptivity[28]. ZYP was found to ameliorate precocious endometrial maturation in a mouse model of different ovarian stimulation[23]. The differential metabolites after ZYP administration may link to the amelioration of endometrial receptivity and oocyte/embryo quality[29]. ZYP demonstrated beneficial effects in patients with diminished ovarian reserve, increasing high-quality embryos partly due to the up-regulation of BMP15 and GDF9 expressions[25]. Several constituents of ZYP showed potential benefits for ovarian function and endometrial receptivity in preclinical studies. Among these, herbs such as Panax ginseng and Lycium barbarum

exhibited the ability to improve reproductive capacity in aged models[30,31]. Panax ginseng, Rehmanniae Radix Praeparata, and Eucommia Cortex exerted ameliorative effects on ovarian function[32–34]. Semen Cuscutae demonstrated potential in promoting embryo implantation[35]. Of note, a previous study investigated the relationship between the chemical components of ZYP and their pharmacological efficacy using the endometrial receptivity disorder mouse model and premature ovarian failure mouse model[36]. However, the precise interactions among the multiple herbal constituents and their combined effects on reproductive outcomes remain largely unresolved, highlighting the need for further mechanistic research.

To our knowledge, the current study was the first large-scale, double-blind, placebo-controlled, RCT to assess the effect of oral preparation on live birth in the context of ART among the AMA

**Table 1 | Characteristics of the participants at baseline**

| Characteristics | ZYP group (n = 734) | Placebo group (n = 733) |
|---|---|---|
| Age, years, median (IQR range) | 37.0 (36.0–39.0) | 37.0 (36.0–39.0) |
| Weight, kilogram, median (IQR range) | 58.0 (52.0–63.0) | 57.0 (52.0–62.0) |
| Height, centimeter, median (IQR range) | 160.0 (156.0–163.0) | 159.0 (156.0–163.0) |
| BMI, kg/m$^2$ | 22.72 ± 2.61 | 22.66 ± 2.64 |
| Type of infertility | | |
| Primary infertility | 181 (24.7) | 171 (23.3) |
| Secondary infertility | 545 (74.3) | 554 (75.6) |
| Duration of attempt to conceive, years, median (IQR range) | 3.0 (2.0–6.5) | 4.0 (2.0–7.0) |
| Concomitant infertility factors | | |
| Pelvic factors and tubal factors | 519 (70.7) | 542 (73.9) |
| Endometriosis | 22 (3.0) | 13 (1.8) |
| Male factors | 331 (45.1) | 345 (47.1) |
| Unexplained factors | 35 (4.8) | 25 (3.4) |
| Ovulation disorder | 78 (10.6) | 96 (13.1) |
| Scar uterus | 88 (12.0) | 93 (12.7) |
| Diabetes | 32 (4.4) | 35 (4.8) |
| Obstetric History | | |
| Gravidity | 2 (1, 3) | 2 (1, 3) |
| Parity | 1 (0, 1) | 1 (0, 1) |
| Abortion history | 0 (0, 1) | 1 (0, 1) |
| Previous IVF cycles | | |
| 0 | 604 (83.4) | 591 (81.7) |
| 1 | 54 (7.5) | 71 (9.8) |
| 2 | 48 (6.6) | 49 (6.8) |
| ≥3 | 18 (2.5) | 12 (1.7) |
| Previous miscarriage | 138 (18.8) | 148 (20.2) |
| AFC, no., median (IQR range) | 11 (7–16) | 11 (8–15) |
| Basal sex hormones, median (IQR range) | | |
| AMH (ng/mL) | 2.10 (1.29–3.69) | 2.06 (1.16–3.36) |
| E$_2$ (pmol/L) | 160.74 (113.99–238.21) | 159.61 (110.10–248.94) |
| FSH (IU/L) | 6.83 (5.44–8.64) | 6.84 (5.31–8.49) |
| LH (IU/L) | 4.05 (2.89–5.36) | 4.09 (3.01–5.50) |
| PRL (mIU/L) | 281.85 (207.81–384.67) | 283.66 (198.75–387.91) |
| T (nmol/L) | 0.90 (0.59–1.37) | 0.94 (0.59–1.38) |

Categorical variables were summarized with frequencies and percentages. Normally distributed variables were presented as means and standard deviation (SD), Non-normally distributed variables were presented as medians and interquartile ranges (IQR). *ZYP* Zishen Yutai Pill; *IQR* interquartile range; *SD* standard deviation; *BMI* body mass index; *AFC* antral follicular count; *IVF* in vitro fertilization; *AMH* anti-Müllerian hormone; *E$_2$* estradiol; *FSH* follicular-stimulating hormone; *LH* luteinizing hormone; *PRL* prolactin; *T* testosterone.

population. According to the statistics from the International Committee for Monitoring Assisted Reproductive Technologies, approximately 3 million cycles were performed annually in recent years, with half being initiated as fresh IVF/ICSI cycles[37]. One-third of these cycles are performed among AMA women aged 35–39[3,38]. The effectiveness of ZYP − an absolute difference of 5.3% in ITT analysis (Supplementary Table 1) − translates to approximately 26,500 additional live births globally each year. As China accounts for roughly one-third of all embryo transfers worldwide, the positive effects of ZYP could potentially lead to approximately 8800 additional live births annually in China alone, which corresponds to around 10,000 neonates each year.

In the field of ART, even modest improvements in LBR can have profound clinical significance, with a 5–10% increase widely recognized as the minimal clinically important difference[39]. While our ITT analysis showed a 4.3% absolute increase in live birth rates, it is important to note that LBR in the placebo group was only 19%. A 22.6% relative increase could be critical to the AMA women, for whom each missed opportunity may represent a diminishing chance of achieving motherhood as time progresses[5]. In addition, both the per-protocol analysis (41.0% vs. 33.7%, $P = 0.040$) and analysis among women with embryo transfer (40.5% vs. 33.7%, $P = 0.043$) revealed a 7% absolute increase in LBR. This effect size is comparable to some landmark studies in reproduction medicine studies, like the novel agent OXO-001 (7% live birth rate increase, analysis among women with embryo transfer), and fresh embryo transfer vs. the freeze-all strategy in women with low prognosis for IVF treatment (8% absolute difference in LBR favoring fresh embryo transfer)[40,41]. These studies align with our findings that even single-digit absolute improvements (4–7%) represent clinical importance in ART. Unlike antioxidant supplements, which lack robust evidence, ZYP −as a non-invasive oral therapy−achieves effectiveness comparable to new drugs with a well-established safety profile and high accessibility[20,42].

## Limitation

Several limitations should be considered in the current trial. First, the trial exclusively evaluated fresh embryo transfer cycles. Given the increasing prevalence of frozen-thawed cycles in contemporary ART practice globally, the generalizability of the observed live birth rate increase to AMA women in current practice settings remains uncertain.

**Table 2 | Outcomes of ovarian stimulation and characteristics of embryo transfer**

| Characteristics | ZYP group (n = 734) | Placebo group (n = 733) |
|---|---|---|
| Protocol | | |
| Long protocol | 518 (70.6) | 507 (69.2) |
| Antagonist protocol | 193 (26.3) | 199 (27.1) |
| Days of ovarian stimulation, days, median (IQR range) | 11 (9–13) | 11 (9–13) |
| Gonadotropin dose, IU, median (IQR range) [a] | 1575.00 (656.25–2612.50) | 1500.00 (515.63–2475.00) |
| Estradiol on hCG trigger day, pmol/L, median (IQR range) | 8826.35 (5794.93–13990.04) | 9446.58 (5449.95–14441.45) |
| Progesterone on hCG trigger day, μg/L, median (IQR range) | 0.70 (0.48–1.05) | 0.70 (0.45–1.04) |
| Luteinizing hormone on hCG trigger day, IU/L, median (IQR range) | 1.70 (1.05–2.64) | 1.55 (1.04–2.59) |
| Endometrial thickness on hCG trigger day, mm, median (IQR range) | 11.00 (9.50–12.80) | 11.00 (9.25–12.80) |
| Outcomes | | |
| Number of oocytes retrieved, no., median (IQR range) | 9 (5, 13) | 9 (6, 13) |
| Number of cleavage, no., median (IQR range) | 7 (4, 10) | 6 (4, 10) |
| Number of 2PN fertilization, no., median (IQR range) | 6 (3, 9) | 6 (3, 9) |
| Number of available embryos, no., median (IQR range) | 4 (2, 5) | 3 (2, 5) |
| Number of high-quality embryos, no., median (IQR range) | 2 (0, 4) | 2 (0, 4) |
| Method of fertilization [b] | | |
| IVF | 324 (76.8) | 318 (77.2) |
| ICSI | 75 (17.8) | 63 (15.3) |
| IVF + ICSI [c] | 23 (5.5) | 31 (7.5) |
| Number of embryos transferred [b] | | |
| 1 | 148 (35.1) | 158 (38.3) |
| 2 | 270 (64.0) | 253 (61.4) |
| 3 | 4 (0.9) | 1 (0.2) |
| The day of embryo-transfer [b] | | |
| D2 | 5 (1.2) | 2 (0.5) |
| D3 | 319 (75.6) | 320 (77.7) |
| D4 | 2 (0.5) | 4 (1.0) |
| D5 | 96 (22.7) | 86 (20.9) |

Categorical variables were summarized with frequencies and percentages. Normally distributed variables were presented as means and standard deviation (SD), non-normally distributed variables were presented as medians and interquartile ranges (IQR). *ZYP* Zishen Yutai Pill; *IQR*, interquartile; *SD* standard deviation; *hCG* human chorionic gonadotropin; *IVF* in vitro fertilization; *ICSI* intracytoplasmic sperm injection (IVF/ICSI).
[a]Gonadotrophin dose refered to the total dose during ovarian stimulation.
[b]Denominators were participants who underwent embryo transfer, 422 in the ZYP group and 412 in the placebo group.
[c]"IVF + ICSI" refers to both "IVF" and "ICSI" were applied during fertilization. The combination of IVF and ICSI was used for patients at risk of low or no fertilization with conventional IVF to mitigate this risk. This included women with unexplained infertility for 7 years or longer, as well as those whose husbands had a history of significant fluctuations in sperm concentration or motility.

Second, the study did not assess cumulative live birth outcomes, lacking a comparison of treatment effectiveness over multiple cycles. Third, in our study, the number of mature oocytes for IVF cycles was retrospectively defined based on day 1 pronuclear and second polar body counts, rather than at the time of retrieval. Although this approach was prespecified in the protocol (Supplementary Note 1), it deviates from the current consensus that maturity should be assessed at retrieval[43]. To avoid potential misinterpretation, we therefore removed this outcome from our results. Fourth, this trial used stratified randomization by age but did not include the stratification variable in the prespecified primary analysis model. Our primary analysis was a simple comparison of LBR between treatment groups. Although this approach followed the original statistical analysis plan, it differs from a recent recommendation suggesting covariate adjustment for stratification variables[44]. To address this, we performed a post hoc logistic regression analysis, which yielded results consistent with the primary analysis, supporting the robustness of our findings (see Supplementary Table 11). Fifth, the trial was powered at 80% for the trade-off between feasibility and precision. There remains a 20% risk of type II error.

ZYP may cause mild side effects (e.g., GI discomfort, rash, dry mouth), but these overlap with symptoms of ovarian stimulation. Adverse events were systematically recorded (Supplementary Table 6), and no clear pattern of unblinding was observed. While unblinding cannot be fully excluded, the likelihood appears minimal.

Furthermore, as this trial involved participants exclusively from China, external validation is necessary to determine applicability to other ethnic or geographic populations. We emphasized that these findings must be interpreted by qualified professionals within appropriate clinical or scientific contexts to avoid misapplication. Key considerations include the specific characteristics of our study population (Chinese ethnicity, BMI < 28 kg/m², aged 35–42) and IVF protocol constraints (evaluation limited to GnRH-a long protocol or GnRH-ant protocol, with fresh-cycle embryo transfer only).

## Methods
### Study design
This multicenter, prospective, double-blind, placebo-controlled, randomized clinical trial (RCT) was done at 12 tertiary-level hospitals in China. The trial protocol with two major revisions (Supplementary Note 1 and 2) received approval from the ethics committees of all sites. The first revision (approved on September 20, 2018) updated the list of participating sites, modified the BMI inclusion criterion, sample size and collection of semen parameters. The second revision (approved on May 30, 2019) updated the list of participating sites, revised the primary outcome and refined the definition of high-quality embryos. In

**Table 3 | Treatment outcomes in intention-to-treat analyses**

| Outcome | ZYP group (n = 734) | Placebo group (n = 733) | Relative Ratio (95% CI) | Absolute Difference between Groups (95% CI) | P |
|---|---|---|---|---|---|
| **Primary outcome** | | | | | |
| Live birth, n (%) | 171 (23.3) | 139 (19.0) | 1.23 (1.01 ~ 1.50) | 4.3 (0.2 ~ 8.5) | 0.042 |
| Singleton | 145 (19.8) | 117 (16.0) | 1.24 (0.99 ~ 1.54) | 3.8 (− 0.1 ~ 7.7) | 0.058 |
| Twin | 26 (3.5) | 22 (3.0) | 1.18 (0.68 ~ 2.06) | 0.5 (−1.3 ~ 2.4) | 0.560 |
| **Secondary outcomes** | | | | | |
| Biochemical pregnancy, n (%) | 232 (31.6) | 195 (26.6) | 1.19 (1.01 ~ 1.40) | 5.0 (0.4 ~ 9.6) | 0.035 |
| Implantation rate, no./ total no. (%) | 252/700 (36.0) | 204/667 (30.6) | 1.18 (1.01 ~ 1.37) | 5.4 (0.4 ~ 10.4) | 0.034 |
| Clinical pregnancy, n (%) | 216 (29.4) | 177 (24.1) | 1.22 (1.03 ~ 1.45) | 5.3 (0.8 ~ 9.8) | 0.022 |
| Pregnancy loss, no./total no. (%) | | | | | |
| Among biochemical pregnancy | 57/232 (24.6) | 51/195 (26.2) | 0.94 (0.68 ~ 1.30) | −1.6 (− 9.9 ~ 6.6) | 0.707 |
| Among clinical pregnancy | 45/216 (20.8) | 38/177 (21.5) | 0.97 (0.66 ~ 1.42) | − 0.6 (− 8.9 ~ 7.4) | 0.878 |
| First trimester | 39/216 (18.1) | 34/177 (19.2) | 0.94 (0.62 ~ 1.42) | −1.2 (− 9.0 ~ 6.5) | 0.770 |
| Second trimester | 6/216 (2.8) | 4/177 (2.3) | 1.23 (0.35 ~ 4.29) | 0.5 (− 3.2 ~ 4.0) | 1.000 |
| Birth weight, g, median (IQR range) | | | | | |
| Singleton | 3250.0 (2980.0–3545.0) | 3260.0 (3000.0–3600.0) | NA | − 30.5 (− 151.0 ~ 89.9) | 0.625 |
| Twin | 2450.0 (2150.0–2600.0) | 2275.0 (2085.0–2595.0) | NA | 54.8 (− 142.8 ~ 252.4) | 0.417 |
| Gestation age, week, median (IQR range) | | | | | |
| Singleton | 39.0 (38.0–39.3) | 39.0 (38.0–39.6) | NA | − 0.27 (-0.71 ~ 0.17) | 0.379 |
| Twin | 36.3 (34.3–37.0) | 36.0 (34.0–37.1) | NA | 0.05 (− 1.45 ~ 1.55) | 0.983 |

Note: As shown in Fig. 1, six pregnancies occurred after randomization but before embryo transfer (ET) in each arm (*n* = 12 in total). These pregnancies were unrelated to the ET procedures and, therefore excluded from the denominator when calculating fresh cycle live birth and other pregnancy-related outcomes.

**Table 4 | Treatment outcomes in per protocol analyses**

| Outcome | ZYP group (n = 376) | Placebo group (n = 374) | Relative Ratio (95% CI) | Absolute Difference between Groups (95% CI) | P |
|---|---|---|---|---|---|
| **Primary outcome** | | | | | |
| Live birth, n (%) | 154 (41.0) | 126 (33.7) | 1.22 (1.01 ~ 1.47) | 7.3 (0.4 ~ 14.1) | 0.040 |
| Singleton | 131 (34.8) | 105 (28.1) | 1.24 (1.00 ~ 1.54) | 6.8 (0.1 ~ 13.3) | 0.046 |
| Twin | 23 (6.1) | 21 (5.6) | 1.09 (0.61 ~ 1.93) | 0.5 (− 3.0 ~ 4.0) | 0.770 |
| **Secondary outcomes** | | | | | |
| Biochemical pregnancy, n (%) | 211 (56.1) | 180 (48.1) | 1.17 (1.02 ~ 1.34) | 8.0 (0.8 ~ 15.0) | 0.029 |
| Implantation rate, no./ total no. (%) | 228/619 (36.8) | 188/612 (30.7) | 1.20 (1.02 ~ 1.40) | 6.1 (0.8 ~ 11.4) | 0.023 |
| Clinical pregnancy, n (%) | 195 (51.9) | 162 (43.3) | 1.20 (1.03 ~ 1.39) | 8.6 (1.4 ~ 15.6) | 0.019 |
| Pregnancy loss, no./total no. (%) | | | | | |
| Among biochemical pregnancy | 53/211 (25.1) | 49/180 (27.2) | 0.92 (0.66 ~ 1.29) | − 2.1 (− 10.9 ~ 6.6) | 0.637 |
| Among clinical pregnancy | 41/195 (21.0) | 36/162 (22.2) | 0.95 (0.64 ~ 1.41) | −1.2 (− 9.9 ~ 7.3) | 0.784 |
| First trimester | 35/195 (17.9) | 32/162 (19.8) | 0.91 (0.59 ~ 1.40) | −1.8 (− 10.1 ~ 6.3) | 0.664 |
| Second trimester | 6/195 (3.1) | 4/162 (2.5) | 1.25 (0.36 ~ 4.34) | 0.6 (− 3.5 ~ 4.4) | 1.000 |
| Birth weight, g, median (IQR range) | | | | | |
| Singleton | 3280.0 (2980.0–3540.0) | 3240.0 (3000.0–3550.0) | NA | 19.7 (− 106.0 ~ 145.3) | 0.818 |
| Twin | 2450.0 (2150.0–2600.0) | 2275.0 (2080.0–2600.0) | NA | 62.5 (− 144.2 ~ 269.3) | 0.427 |
| Gestation age, week, median (IQR range) | | | | | |
| Singleton | 39.0 (38.0–39.1) | 39.0 (38.0–39.6) | NA | − 0.25 (− 0.70 ~ 0.20) | 0.328 |
| Twin | 36.9 (34.6–37.0) | 36.0 (34.0–37.0) | NA | 0.24 (− 1.35 ~ 1.84) | 0.643 |

Per-protocol (PP) analysis was performed, excluding those who had major protocol deviation(s), canceled cycle or did not complete the pre-set minimum exposure dosage of the assigned study drug (at least 80% compliance), from the ITT population.

this revision, the clinical pregnancy rate was removed as one of the two primary outcomes, and LBR was left as the only primary outcome, because it is the most clinically relevant endpoint in ART trials. Notably, the sample size calculation was solely based on LBR. The specification of "fresh cycle" was added to avoid ambiguity and ensure consistency in analysis (Supplementary Note 2).

Written informed consent was obtained from all participants (Supplementary Note 3).

Lead approval was granted by Ethics Committee of Reproductive Medicine, Sun Yat-sen Memorial Hospital of Sun Yat-sen University (2017 Reproduction Ethnic Approval No.2, in Chinese, 2017 生殖伦审字第(02)号). Ethic approvals were also gained in all other participating sites, including Clinical Application and Ethics Committee of Human Assisted Reproductive Technology of Women and Children's Hospital of Chongqing Medical University; IEC of Institution for National Drug Clinical Trials of Tangdu Hospital, Fourth Military Medical University; The Research and Clinical Trial Ethics Committee of the First Affiliated Hospital of Zhengzhou University; Ethics Committee of First Hospital of Lanzhou University; Ethical Committee for Drug Clinical Trials of Liuzhou Maternity and Child Healthcare Hospital; Medical Ethics Committee of West China Second University Hospital, Sichuan University; The Ethics Committee of the Affiliated Suzhou Hospital of Nanjing Medical University; Clinical Application Ethics Committee of Human Assisted Reproduction Technology of Northwest Women and Children's Hospital; Ethics Committee of Reproductive and Genetic Hospital of CITIC-Xiangya; Ethics Committee for Drug Clinical Trials of the Third Affiliated Hospital of Zhengzhou University; Ethics Committee of Hospital for Reproductive Medicine Affiliated to Shandong University. The names of all ethics committees and the reference number of ethics approval were available in Supplementary Note 4. The trial aimed to compare the fresh cycle LBR in AMA women after IVF/ICSI-ET.

## Participants

The following inclusion and exclusion criteria were based on our previous research, but with differences in age and body mass index (BMI) compared to our previous studies[20].

The trial included infertile women planning to undergo IVF/ICSI-ET, aged 35–42 years, with a body mass index (BMI) below 28 kg/m², who had both ovaries present and provided written informed consent.

Women were excluded if they had recurrent implantation failure (three or more IVF/ICSI-ET failures); had adenomyosis, or submucosal fibroids/intramural fibroids that caused distortion of the endometrial cavity; had untreated bilateral hydrosalpinx; had active or unresolved endometrial disease (e.g., endometrial polyp, chronic endometritis, intrauterine adhesions) or other diseases not suitable for ART or pregnancy; had received TCM therapy for infertility within one month.

All participants and their husbands gave written informed consent before enrollment. We followed the CONSORT extension reporting guidelines for Chinese herbal medicine formulations. The trial was registered at ClinicalTrials.gov, NCT03703700.

## Randomization and masking

Randomization was performed prior to the start of the ovarian stimulation cycle (see Supplementary Note 1). Participants were randomly assigned in a 1:1 ratio to either the ZYP or placebo arm using the permuted block randomization method, with a fixed block size of 4. The randomization was stratified by the age group (35–37, 38-39 and 40–42) in line with categories described by the Human Fertilization and Embryology Authority[45]. Allocation concealment was ensured through a centralized, web-based Interactive Response Technology (IRT) system managed by an independent third-party company (Guangzhou Evidence-Based Medicine Tech Co., Ltd), which also handled randomization, eCRF maintenance, and drug dispensing. Study personnel (investigators, clinical staff, and participants) had no

access to the randomization sequence. Enrollment was performed by clinical investigators at participating centers, who had no role in the allocation process.

Both ZYP and placebo were manufactured and provided by Guangzhou Baiyunshan Zhongyi Pharmaceutical Co. Ltd. The ZYP contains 15 herbal drugs (Supplementary Note 5). The manufacturer of ZYP complies with the relevant requirements of the law of China's Drug Administration and Good Manufacturing Practice (GMP), with approval from the China National Medical Products Administration (Permit No. Z44020008). ZYP exhibited a similarity of more than 0.82 by ultrahigh performance liquid chromatography-charged aerosol detector (UPLC-CAD), indicating the ingredients of ZYP are stable and controllable according to the quality control method. Detailed quality control protocols and results were provided in Supplementary Note 5[46]. All intervention drugs were identical in packaging, appearance, and odor (see Supplementary Note 5).

In our protocol, unblinding was permitted in the event of a medical emergency. However, throughout the whole trial, blinding remained intact as no instances of such emergencies occurred. The allocation was disclosed by the statistician managing the IRT after the retrieval of all live birth data. A data safety and monitoring board oversaw the trial (Supplementary Note 1).

## Procedures

All participants received standardized OS protocols, including gonadotropin-releasing hormone agonist (GnRH-a) long protocol or GnRH-antagonist (GnRH-ant) protocol.

1) GnRH-a long protocol: Participants in both arms underwent OS after down-regulation with long-acting or short-acting GnRH-a in the previous midluteal phase. Hormone levels, including follicle-stimulating hormone (FSH), luteinizing hormone (LH) level and estradiol ($E_2$) were measured on the gonadotropin (Gn) initiation day. Follicular development (diameter and counts) and endometrial thickness were evaluated by means of transvaginal ultrasonography. OS was initiated when the serum LH < 5 IU/L and $E_2$ < 50 ng/ml or endometrial thickness < 5 mm.

2) GnRH-ant protocol: Participants in both arms underwent initiation on day 2 to day 4 of the menstrual cycle. Hormone levels, follicular development and endometrial thickness were also monitored as described in the GnRH-a long protocol.

The initiation dose and total Gn dose were adjusted according to a combined consideration of age, weight, basal hormone levels, AFC and previous ovarian response. The mean diameter of all follicles and endometrial thickness were monitored by means of transvaginal ultrasonography, along with testing of serum hormonal levels, including FSH, LH, $E_2$ and progesterone ($P_4$) on day 5 to day 6 after Gn initiation. When the follicular size met the criteria (two leading follicles ≥ 18 mm; or ≥ 3 follicles ≥ 17 mm; or ≥ 4 follicles ≥ 16 mm), ovulation was triggered by injecting human chorionic gonadotropin (HCG). Serum FSH, LH, $E_2$ and $P_4$ levels were monitored on the HCG injection day. Oocyte collection was conducted 36 h later. Luteal support commenced after oocyte retrieval. Fertilization was achieved by either IVF or ICSI with the husband's semen. After oocyte retrieval, embryo transfer was performed on day 3 - 5 as appropriate.

The intervention flowchart was presented in Supplementary Note 1. Briefly, study drug intervention, ZYP or placebo, was administered 3 times daily, 5 g each time, from down-regulation day (GnRH-a long protocol) or day 19 to day 23 of the previous menstrual cycle (GnRH-ant protocol), until 2 weeks after ET. Study drug discontinued during the 1st day to the 4th day of the menstrual cycle. The study drug intervention would continue when a biochemical pregnancy was confirmed as positive. Intervention of the study drug stopped in the case of a negative pregnancy test. For patients achieved positive results in the pregnancy test, the study drugs were administered until

the confirmation of clinical pregnancy (five weeks after ET). This treatment protocol was based on the instructions of the ZYP manufacturer and used in a previous study.

Five visits were planned in the protocol (Supplementary Note 1). Delivery and neonate information was retrieved by investigators through web or telephone contact.

## Outcomes

Following our previous report, live birth rate was the primary outcome[20]. Live birth was defined as the delivery of any viable infants after 28 weeks of gestation in a fresh embryo transfer cycle.

The secondary outcomes included counts and rates of oocytes/embryos (oocytes retrieved, 2 pro-nuclei zygotes, cleavage zygotes, available embryos, high-quality embryos), pregnancy outcomes (rates of biochemical pregnancy, implantation, clinical pregnancy and miscarriage), incidences of maternal, fetal and neonatal complications, and neonatal information (newborn birth weight, length, congenital malformations).

A high-quality embryo was defined according to the day of embryo transfer, following the Istanbul consensus and Gardner criteria, Day 2: 4 cells, cell fragments < 10% and no multi-nucleus; Day 3: 8 cells, cell fragments < 10%, no multi-nucleus; Day 5: stage 4 blastocyst, grade A inner cell mass, grade A trophectoderm[47,48].

Biochemical pregnancy was defined as a serum β-hCG level of > 50 IU/L. Implantation rate was defined as the number of gestational sacs per number of embryos transferred. Clinical pregnancy was defined as the presence of an intrauterine gestation sac with detectable fetal cardiac activity under transvaginal ultrasonography. The miscarriage rate was analyzed among patients with positive pregnancy tests and patients with clinical pregnancies. Detailed definitions of maternal, fetal and neonatal complications were listed in Supplementary Note 1.

## Statistical analysis

In the field of reproductive medicine, a plausible and clinically meaningful improvement of LBR is generally considered to be 5-10%[39]. In our previous clinical research, the LBR was 0.42 per embryo transfer in ZYP treated group and 0.33 per embryo transfer in the placebo control group among AMA women, which aligned with this improvement and provided an empirically derived effect size of ZYP treatment[21]. Thus, we performed sample size estimation based on this effect size. For feasibility, we chose a power of 80%. To detect this difference with 80% power at a two-sided significance level of 0.05, we estimated that 454 participants per arm who actually underwent embryo transfer would be required. According to our previous study, the proportion of participants who did not reach embryo transfer due to dropout (15%) or cycle cancellation (23%), resulting in a combined attrition rate of approximately 38%. While computing the sample size for this trial, we considered a total rate of 38% (dropout and cancellation). Therefore, we calculated that at least 733 participants in each group were required.

The primary analysis was performed according to the intention-to-treat (ITT) principle, i.e., analyzing all randomized participants in their originally assigned groups. Prior to unblinding, missing value in the biochemical pregnancy, clinical pregnancy and live birth in all analyses were taken as not having an event.

Categorical variables were summarized with frequencies, percentages, relative ratio (RR) and absolute difference of proportion with 95% confidence interval (CI). Between group comparisons were performed with $\chi^2$ test or Fisher's exact test, as appropriate. For RR, the Delta method was used to calculate the 95% CI. For the absolute difference in proportions, the Newcombe method was employed to calculate 95% CI. For continuous variables, the distributions were assessed with the Kolmogorov-Smirnov test. Normally distributed variables were presented as means and standard deviation (SD), assessing intergroup differences with $t$ test. Non-normally distributed variables were presented as medians and interquartile ranges (IQR), assessing intergroup differences with the Mann-Whitney U test.

Per-protocol (PP) analysis was performed, excluding those who had major protocol deviation(s), canceled cycle or did not complete the pre-set minimum exposure dosage of the assigned study drug (at least 80% compliance), from the ITT population.

Post hoc analyses were also conducted, among the population with embryo transfer, comparing the pregnancy outcomes per embryo transfer. A pre-defined subgroup analysis was conducted as stratified by age group (35–37, 38-39, 40–42). Due to the similarity of our two RCT, an exploratory analysis was also conducted, pooling pregnancy data from the two databases[21].

SPSS 19.0 was applied in statistical analyses. Two-sided $P$-value < 0.05 was considered as significantly different.

## Reporting summary

Further information on research design is available in the Nature Portfolio Reporting Summary linked to this article.

## Data availability

De-identified individual participant data are available at https://github.com/hepingzhangyale/Zishen-Yutai-Pill. Additional information and results related to the trial can be found at clinicaltrials.gov under accession: NCT03703700 (https://www.clinicaltrials.gov/). Study protocol, statistical analysis plan and informed consent form are available in Supplement Files.

## Code availability

The code used to analyze the data in the paper can be found in https://github.com/hepingzhangyale/Zishen-Yutai-Pill.

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

## Acknowledgements

This study was supported by the Guangdong Provincial Bureau of Traditional Chinese Medicine (Project No: 20174002), Science and Technology Program of Guangzhou, China (Project No: 201704020046) through the assistance of Dongzi Yang and Yu Li (Supplementary Note 6). The funders had no role in considering the study design or in the collection, analysis, interpretation of data, writing of the report, or decision to submit the article for publication. Quality control protocols were provided by the Guangzhou Baiyunshan Zhongyi Pharmaceutical Co. Ltd. and included in Supplementary Note 5. We thank all the members in Data and Safety Monitoring Board for their help in coordination during the trial, including Zhaosi Xu (Guilin University of Electronic Technology), Xiaoli Chen (Sun Yat-Sen Memorial Hospital of Sun Yat-Sen University), Jiewen Zhou (Guangzhou University of Chinese Medicine).

## Author contributions

H.Z. and D.Y. supervised the study and contributed equally as corresponding authors. Y.L., H.Z., and D.Y. were responsible for the study conception and design, and wrote the study protocol. Y.L. and D.Y. were responsible for funding acquisition. Y.L., F.G., X.W., L.H., H.L., Y.G., H.Y., L.F., H.B., Z.W., W.H., X.M., and D.Y. were involved in the recruitment of patients and acquisition of the data. Y.L. drafted the manuscript. H.Z. performed the statistical analysis. All authors involved in the interpretation of data and critical revision of the manuscript for important intellectual content. All authors read and approved the final draft of the report.

## Competing interests

The Authors declare the following competing interests. D.Y. reported that she received grants from Guangzhou Baiyunshan Zhongyi Pharmaceutical Co., Ltd. and The Secondary Development Project of Famous and High-quality Chinese Patent Medicine in Guangdong Province (Project No.: 20174002) during the conduct of the study. Y.L. reported that she received grants from the Science and Technology Program of Guangzhou (Project No.: 201704020046) during the conduct of the study. The remaining authors report no competing interests. Guangzhou Baiyunshan Zhongyi Pharmaceutical Co., Ltd. donated the study drugs, including Zishen Yutai Pill and the placebo, for this research. An independent third-party contract research organization (CRO), Guangzhou Evidence-Based Medicine Tech Co., Ltd., was hired by Guangzhou Baiyunshan Zhongyi Pharmaceutical Co., Ltd. for data management. Guangzhou Baiyunshan Zhongyi Pharmaceutical Co., Ltd had no role in the data acquisition, statistical analysis and writing of this article.

## Additional information

¹Center for Reproductive Medicine, Sun Yat-Sen Memorial Hospital of Sun Yat-Sen University, Guangzhou, Guangdong, China. ²Reproductive Medicine Center, the University of Hong Kong, Shenzhen Hospital, Shenzhen, Guangdong, China. ³Center of Reproductive Medicine, Reproductive and Genetic Hospital of CITIC-Xiangya, Changsha, Hunan, China. ⁴Reproductive Medical Center, Tangdu Hospital, the Fourth Military Medical University, Xi'An, Shanxi, China. ⁵Center for Reproductive Medicine, The First Affiliated Hospital of Zhengzhou University, Zhengzhou, Henan, China. ⁶Reproductive Genetic Center, The Affiliated Suzhou Hospital of Nanjing Medical University, Suzhou, Jiangsu, China. ⁷Reproductive Medicine Center, The Third Affiliated Hospital of Zhengzhou University, Zhengzhou, Henan, China. ⁸Center for Reproductive Medicine, Women and Children's Hospital of Chongqing Medical University, Chongqing, China. ⁹Department of Reproductive Medicine, Liuzhou Maternity and Child Healthcare Hospital, Liuzhou, Guangxi, China. ¹⁰Assisted Reproduction Center, Northwest Women's and Children's Hospital, Xi'An, Shanxi, China. ¹¹Center for Reproductive Medicine, Hospital for Reproduction Medicine Affiliated to Shandong University, Jinan, Shandong, China. ¹²Division of Reproductive Medical Center, West China Second University Hospital of Sichuan University, Chengdu, Sichuan, China. ¹³Reproductive Medicine Center, The First Hospital of Lanzhou University, Lanzhou, Gansu, China. ¹⁴Department of Biostatistics, Yale University School of Public Health, New Haven, Connecticut, USA. ✉e-mail: yangdz@mail.sysu.edu.cn; heping.zhang@yale.edu

