## [Transparent Peer Review file · Nature Communications]

Zishen Yutai Pill increased live births in advanced maternal age women: A randomized clinical trial

Corresponding Author: Professor Heping Zhang

Version 1:

Reviewer comments:

Reviewer #1

(Remarks to the Author)

This was a well-designed trial and is well reported in accordance with the CONSORT guidelines. However, there are a few, mostly fixable flaws, that need to be addressed to strengthen the manuscript.

1. Abstract

- a. In the first paragraph write ART in full as it's the first time the abbreviation is used
- b. Line 6..... assigned in 1:1 ratio to received ZYP: please change received to receive

2. Background: The authors have presented a good background to the study, however, a statement of specific objectives, both primary and secondary, is missing at the end of the introduction section, hence needs to be included. There is only a single statement on the aim of the study, but no specific objectives.

3. Methods

- a. Could the authors please provide more information on randomisation including the number of block and the methods to ensure allocation concealment, who enrolled participants
- b. There was only one stratification variable, age, used in the trial. Could the authors provide an assurance that there was good between-arm balance within the recruitment centres (12 tertiary-level hospitals) as centre was not a stratification variable?
- c. Sample size: the effect size/detectable difference is based on the observed difference in a previous study rather than a minimally clinically important difference, could the authors explain this? Also, the authors need to justify the choice of 80% power as opposed to 90%.

4. Statistical analysis

- a. "Intention to treat" is ambiguous so authors need to clearly define the primary analysis set and the per-protocol set detailing who was excluded in the per-protocol set. Some information about this can be deduced from the results section in the CONSORT diagram, but this needs to be explicitly described.
- b. A stratification variable was used in the randomisation. According to many published guidance documents stratification variables should usually be included as covariates in the primary analysis. The presented analysis is unadjusted so this needs to be amended to reflect the recommended practice. Also, it is unclear how the confidence intervals were obtained for some of the univariable analyses reported such as the Chi-squared tests for the categorical variables.
- c. Handling of missing data: there are principled ways of dealing with missing data, but the authors seemed to have used methods such as best-case scenario replacing missing data with not having an event. Could the authors justify this and in addition perform some sensitivity analyses on missing data.

5. Results

- a. CONSORT diagram: a small typo- in the Figure 1. Study flowchart, in the Box 2, please change" 27 Not fulfilling the eligible criteria" to "27 Not fulfilling the eligibility criteria"
- b. Baseline table: Some characteristics e.g. age, weight, height are almost identical between group which is not typical in real life RCTs. This is too good to be real, so could the authors verify that this is correct. Also, some of the reported standard deviation like SD for age seem to be quite small so the authors need to verify.
- c. Duration of attempt to conceive shows a median difference between groups of 1 year which is quite a long time. How was this variable recorded- in years or months?
- d. LBRs are quite different between the ITT and the per-protocol analyses- could the authors explain the implication of this?
- e. Could the authors relate the observed between group difference in the primary analysis and the assumed difference in the sample size calculation and discuss the implications in terms of study power?

Reviewer #2

(Remarks to the Author)

The manuscript has been carefully examined. The study reported a multicenter, prospective, double-blind, placebo controlled, randomized trial (NCT03703700) aims to evaluate the effectiveness of Zishen Yutai Pill (ZYP), a SFDA approved TCM formula for increasing live births in advanced maternal age (AMA) women with infertile. This study involved total 1467 AMA infertile women participants (734 in ZYP arm, 733 in placebo arm) from 12 tertiary level hospitals in China. The subjects were randomly assigned in 1:1 ratio to receive oral treatment of ZYP or placebo (5 g once, three times daily) around the time of embryo transfer. The primary outcome was live birth. ZYP treatment group had live birth occurred in 23.3% (171 of 734) whereas the placebo group 19.0% (139 of 733) (relative ratio, 1.23 [95%CI, 1.01 to 1.50], showing absolute difference 4.3% [95%CI, 0.2% to 8.5%]; P = 0.042). No significant differences were found regarding controlled ovarian hyperstimulation outcomes, or maternal, fetal and neonate adverse events (all P > 0.05). The study indicates that ZYP treatment could enhance live birth rate during fresh cycles in AMA women undergoing IVF/ICSI-ET. This study provides strong evidence to support ZYP as adjunct therapy with IVF/ICSI-ET to increase the live birth rates. Overall, the study is well designed high quality clinical trials and high impact on increasing the live birth, bringing hope for AMA) women with infertile. Thus, this manuscript can be accepted for publication. However, there are following concerns to be addressed:

1. ZYP was provided 5 g once, three times daily around the time of embryo transfer. What is the rationale of such treatment protocol for the patients? There is not clear information provided regarding to the treatment period of ZYP or placebo.
2. A total of 633 women (43.1%) did not receive embryo transfer or dropped out of the study (312 in the ZYP group and 321 in the placebo group) before ET. The dropping rates are very high. Whether the dropping rates would bring impacts on data interpretation should be justified.
3. Whether the ZYP has potential toxic effects on liver or renal functions? Safety issue is one of the concerns. Basic clinical biochemistry data should be provided.
4. There is no previous marriage history and gravidity in the manuscript.
5. With many herbal items in the ZYP, the authors should have discussion about the potential interactions of different herbal items in promoting live birth.
6. Given that the papers published in Nature Communication are open accessed, cautious information should be provided to prevent the potential misuse of the information

Reviewer #3

(Remarks to the Author)

Reviewer #4

(Remarks to the Author)

Zishen Yutai Pill review

The authors have conducted a multicenter, prospective, double-blind, placebo controlled, randomized trial to assess the effects of Zishen Yutai Pill on clinical outcomes in women of advanced maternal age undergoing fresh embryo transfer. The lack of proposed mechanism of action of the intervention or any features to support an area of mechanism e.g. difference in hormones or endometrial morphology makes it difficult to feel confident about the effects of the intervention. That said it was conducted in a double-blinded manner and a small benefit was found in LBR, although it remains unclear why/how this was achieved.

Abstract-

- The first sentence does not set the scene well for the study. By 'inadequate' do you mean uncertain? As currently implies that it was previously investigated and found to not be effective.
- Population not clearly defined- AMA is not defined in abstract. Other notable inclusion criteria should be presented.
- 'around the time of embryo transfer' is vague.

Introduction

- “Nevertheless, advanced age is conclusively associated with a reduced live birth rate (LBR)” Important to explain the reason for this statement- ie poorer oocyte quality as opposed to reduced implantation, as your intervention seems to be aimed at the latter given that it is applied ‘around the time of embryo transfer’? You have not shown any differences in oocyte / embryo quality nor in endometrial measures?
- “Nevertheless, AMA women had less favorable reproductive tract conditions, altered hormone levels and poorer ovarian reserve compared to the non-AMA women.” – Please specify more clearly what you are referring to?
- How does Zishen Yutai Pill work? What is the biological or scientific basis for the usage of this supplement in women of advanced maternal age?
- “The post hoc subgroup analysis from our previous trial provided promising evidence for the effect of ZYP on AMA women in ART.” Please describe what data this refers to in more detail (samples size and effect size in what parameter) as most readers will not be familiar with your previous trial results. Add reference to your previous trial here also.

Methods

- Should read ‘The protocol is available’
- ‘both ovaries existed’ should read are present.
- ‘by following the Human Fertilisation and Embryology Authority.22’ should read ‘in line with categories described by the HFEA’ or similar.
- Exclusion criteria of: “the uterine cavity line constricted by uterine fibroids”. This is not very clear – does this represent submucosal fibroids, distortion of the cavity by an intramural fibroid, or something else?
- Likewise, in the exclusion criteria: “had uncured endometrial diseases” should be defined more clearly.
- Describe the protocol in more detail. What formulations of FSH were used etc How was the choice of suppression protocol decided and why was it not standardised for all patients? In turn, would this impact the total dose of ZYP administered?
- Why was 28 weeks chosen as the cutoff for live birth outcome as opposed to 24 weeks, which is generally considered to be the cutoff of viability? Please use definitions in line with the CORE outcome set. [https://www.fertstert.org/article/S0015-0282\(20\)32680-7/fulltext](https://www.fertstert.org/article/S0015-0282(20)32680-7/fulltext)
- Clarify that this was live birth per cycle started?
- “Clinical pregnancy was defined as the presence of intrauterine gestation sac with fetal motion under transvaginal ultrasonography” - does this refer to detectable fetal heart activity?
- Under what circumstances was a combination of IVF+ICSI used for fertilization?
- Why was ET performed from D2 – D4 ie non-blastocyst transfer for the majority rather than day 5 blastocyst transfer?
- Can you specify the exact conditions of the Per-Protocol analysis? This is unclear also in Table 4.
- Figure 1 should also describe the suppression protocol split.
- It is still not clear how the investigative medicinal product being investigated is supposed to work? Any useful background information would be useful to describe.
- ‘controlled ovarian hyperstimulation (COH)’ is an outdated term and can now just be described as ovarian stimulation.
- Power- “In our previous clinical research, the LBR was 0.42 per embryo transfer in ZYP treated group and 0.33 per embryo transfer in placebo control group among AMA women.19” State the sample size this statistic is based on. Usually, the power is derived to the ‘minimum clinically important difference’ (MCID) rather than to the expected change in LBR with a particular intervention.

Results

- The reason for not having embryo transfer should be explained more clearly e.g. what is meant by ‘Abnormal endometrium’, and specify level for ‘Elevated E2/P’. What does ‘Unwilling to participate’ mean - as in withdrew consent?
- For completeness, please present comprehensive metrics of the variables in Tables 1 and 2: Min, Mean, SD, Median, Q1, Q3, IQR, Max, N.

- Table 2: is Gonadotrophin dose referring to the total dose during OS?
 - Were patients taking oral ZYP in previous IVF attempts (and how long ago)?
 - Table 2: missing “s” on Number of available embryos.
 - Table 2: the outcome of mature oocytes retrieved is assumingly only relevant to the ICSI cohort? Please specify this as a footnote indicating number of relevant cycles. This is difficult to understand with the "IVF+ICSI" cohort especially.
- Discussion
- Differences in pregnancy loss are insignificant between the two groups which does not align with ZYP being approved for treatment of miscarriage (ref 26)?
 - Did the live birth rate results remain the same when analysis was performed depending on the type of stimulation cycle or stage of embryo transfer ie Cleavage stage only or Blastocyst transfer only?
 - Was the type and dose of luteal phase support administered standardized amongst all patients who received an embryo transfer?
 - Did you observe any differences in oocyte quality or embryo quality in this RCT?
 - “In the field of ART, even modest improvements in LBR can have profound clinical significance, with a 5–10% increase widely recognized as the minimal clinically important difference.^{32, 33}. This argues against the power calculation that you used for this study as much larger numbers would be needed to detect an MCID of 5%.
 - “In addition, both the per-protocol analysis (41.0% vs. 33.7%, P=0.040) and analysis among women with embryo transfer (40.5% vs. 33.7%, P=0.043) revealed a 7% absolute increase in LBR.” How can this be true? Per protocol means including everyone who started the protocol as the denominator ie from 734, so it will be much lower than when using only those who had embryo transfer eg 432.
 - “IVF vs. ICSI in non-severe male factor infertility (4.1% absolute difference in cumulative LBR favoring IVF.^{34, 35, 36}” The difference in LBR in one cycle is not comparable to a cumulative rate.

Other minor comments:

Quality of language can be improved throughout, particularly with errors in usage of tense in all sections apart from the results/discussion.

Abstract- grammatical issues eg:

“Although Zishen Yutai Pill (ZYP) have been demonstrated benefits to women undergoing ART”
 “...its efficacy on the live births of advanced maternal age women remained inadequate”

Intro – issues with tense:

“Advanced maternal age (AMA) is defined as the maternal age of 35 years or older”

“In recent years, the proportion of AMA women undergoing autologous ART had increased from 63.0% in 2014 to 71.8% in 2019 around the globe.”

Page 6 “ZYP had recently been reported to improve the LBR among women undergoing IVF/ICSI.”

Participants:

“both ovaries existed”

Reviewer #5

(Remarks to the Author)

Thank you for the opportunity to review this manuscript which describes the results of a large RCT to evaluate the impact of Zishen Yutai Pill (ZYP) on IVF outcomes in women of advanced age (>35). The study is large and robust, with prospective trial registration and use of a placebo to blind participants and personnel.

I have the following comments and suggestions

Abstract

1. Check the tense as the abstract writes like it's a protocol “aims to evaluate
2. “(5 g once, three times daily)” do you mean 5g three times daily?
3. Abstract should state brief def of AMA (>35)

Intro

4. Please add a description of ZYP to the background including its proposed therapeutic effect. Is it proposed to improve

stimulation response? Embryo quality? Endometrial quality? Etc There is some of this in the Discussion, but readers need it in the intro to understand the study design and question

5. I'm not sure its entirely true that "their effects on pregnancy outcomes, particularly live births, remained unexplored" you will find each of these interventions is covered by a corresponding Cochrane review and that clinical outcomes including pregnancy and live birth are reported in many RCTs – these outcomes are not entirely unexplored

Methods

6. The study was driven by post hoc analysis of an RCT by the same team which showed possible benefit of ZYP for AMA. That study recruited 2014-2017 and was published in 2019. This study recruited from March 4, 2019 to October 14, 2023. However the study appears to have been funded in 2017 based on the funding references provided (20174002 and 201704020046) and the funding agreement provided in the supp files "Project start and end time: January 2017 to December 2019" – please explain.

7. Another possible inconsistency is that recruitment commenced March 2019 but the protocol provided in supp files was written in May 2019. Please explain.

8. The intervention and placebo were manufactured and provided for free by Guangzhou Baiyunshan Zhongyi Pharmaceutical Co. Ltd and this company also hired Guangzhou Evidence-Based Medicine Tech Co. Ltd to do the data collection and randomisation etc. What role did Guangzhou Baiyunshan Zhongyi Pharmaceutical Co. Ltd have in the study protocol and monitoring then? Did the instruct the EBM Tech co in the protocol etc? Their involvement should be clear as they have an obvious COI as do the investigators who receive grant funding from them

9. There are 6 pregnancies resulting in cycle cancellation in each arm per the flow chart – are these included in the total numbers pregnant? Please make it clear.

10. Methods section states "The trial protocol received approval from the ethics committees of all sites" please give the name and reference number for the main ethics approval. I see this is provided elsewhere but it should be clear in the methods

11. Methods – Participants "with modifications in age and body mass index (BMI)." What does this mean? You modified the inclusion criteria during the trial? Or there were modifications compared to your other studies?

12. "permuted block randomization method." – what were the block sizes? Please state

13. When was randomisation carried out? What day in relation to the IVF cycle?

14. The sample size is a little unclear – please provide the expected LBR in each arm after applying the dropout rates etc

Results

15. Per protocol analysis excluded people with cancelled cycles. Is this appropriate? If the ZYP was administered from day 19 it could conceivably impact on the chance of cancellation. Indeed the investigators observed more cancellations in the placebo group. Cancellations for personal reasons may be ok to exclude but I would have expected to include cancellations for poor response/no embryos etc.?

16. There were more multiple embryo transfers in the ZYP group. Why? Could this have contributed to the apparent benefit of ZYP? If appropriate it might be helpful to include the mean number of available and high-quality embryos so we can see if there is any difference, as if not, there is no explanation for more multiple transfer in the ZYP group?

Discussion

17. Discussion states "In the current research, we pooled data from two clinical trials to validate the efficacy of ZYP on fresh cycle live births among AMA women. The results were robust among the ITT population (Table 3), PP population (Table 4) ..." this implies that Table 3 and 4 contain the data from two trials pooled – which I don't understand to be the case. Please rephrase.

18. Discussion states "The ZYP was approved for the treatment of threatened miscarriage and recurrent pregnancy loss" which implies its some sort of regulatory or guideline approval, but the reference is to a meta-analysis. Please check and consider rephrasing.

19. What are the side effects of ZYP? To what extent to the investigators believe the patients and personnel remained blind during the study?

Version 2:

Reviewer comments:

Reviewer #1

(Remarks to the Author)

Please see the attached document

Reviewer #2

(Remarks to the Author)

The authors have fully addressed my comments. I have no further comment on this manuscript. It can be accepted for publication.

Reviewer #3

(Remarks to the Author)

Reviewer #4

(Remarks to the Author)
No further queries.

Reviewer #5

(Remarks to the Author)

Thank you for the opportunity to re-review this manuscript which describes the results of an RCT evaluating ZYP in older women pursuing IVF.

The authors have made numerous substantial changes to the manuscript which has improved transparency and quality of reporting.

I have the following further comments which should be addressed before the manuscript could be accepted

In the response to the previous reviewers comments the authors state that "At first, in the current trial, adverse event monitoring revealed elevated ALT/AST in 2 participants from the ZYP group and 1 from the placebo group. One renal function related adverse event was reported in ZYP group." But I cant see where this info is available in the manuscript or associated supp files? It should be there

The authors state that some protocol changes were made during early days of recruitment, regarding the primary outcome and embryo quality definition. These should be transparently conveyed in the supplementary files/protocol.

Add as a footnote to table 3 that the live births (and presumably all preg outcomes) exclude the 6 per-protocol pregnancies in each arm. Under pure ITT these would be included in the totals so people may be uncertain whether they have been or not.

Regarding blinding – if ZYP has specific side effects, patients can become unblinded despite the techniques used at your centre to conceal the allocations etc. It should be added to the limitations that the study was double-blind and placebo was used, but that it cant be ruled out there was some level of unblinding due to known side effects of ZYP.

I remain concerned that the per protocol analysis approach of excluding those not reaching ET is not the best approach. The authors justify it by a similar approach being used in two studies of fresh vs frozen transfer. This is a different scenario. In this ZYP study we anticipate possible effect of ZYP on the number/quality of embryos (this is probably not expected in the fresh vs frozen example). ZYP could have caused more cancellations in the control group. This is part of how the intervention might work, and not a case of 'violating the protocol'. But as long as the authors clearly explain the inclusions for the analysis, and seeing as it's a secondary analysis anyway, I wont push it further.

Version 3:

Reviewer comments:

Reviewer #1

(Remarks to the Author)

The authors have fully addressed my comments.
I have no further comment on this manuscript.

Reviewer #3

(Remarks to the Author)

Reviewer #5

(Remarks to the Author)

Authors have addressed the outstanding comments and I am happy for this paper to proceed

Dear Reviewers,

Thank you so much for your constructive and insightful suggestions, which were extremely helpful in our effort to revise our manuscript. Our point-by-point responses to your comments are as follows.

Response to Reviewer #1

1. Your comment 1: Abstract-a. In the first paragraph write ART in full as it's the first time the abbreviation is used. b. Line 6..... assigned in 1:1 ratio to received ZYP: please change received to receive

Response: Thank you for your careful review. We apologize for our oversight. Typos and the first uses of all abbreviations are corrected.

2. Your comment 2: Background: The authors have presented a good background to the study, however, a statement of specific objectives, both primary and secondary, is missing at the end of the introduction section, hence needs to be included. There is only a single statement on the aim of the study, but no specific objectives.

Response: The last paragraph in the introduction has been revised to include specific primary and secondary objectives as follows: "Thus, the present study aimed to assess the efficacy and safety of ZYP in enhancing pregnancy outcomes in IVF/ICSI procedures for AMA patients. The primary objective was to evaluate the effect of ZYP on fresh cycle LBR. Secondary objectives included assessing its impact on outcomes of ovarian stimulation (OS), biochemical pregnancy rate, implantation rate, clinical pregnancy rate, miscarriage rate, and occurrence of maternal, fetal and neonatal adverse events."

3. Your comment 3a. Methods: Could the authors please provide more information on randomisation including the number of block and the methods to ensure allocation concealment, who enrolled participants

Response: We have elaborated the randomization and masking in the Methods section as follows: "Participants were randomly assigned in a 1:1 ratio to either the ZYP or

placebo arm using the permuted block randomization method, with a fixed block size of 4. The randomization was stratified by age (35-37, 38-39 and 40-42) in line with categories described by the Human Fertilisation and Embryology Authority. Allocation concealment was ensured through a centralized, web-based Interactive Response Technology (IRT) system managed by an independent third-party company (Guangzhou Evidence-Based Medicine Tech Co., Ltd), which also handled randomization, eCRF maintenance, and drug dispensing. Study personnel (investigators, clinical staff, and participants) had no access to the randomization sequence. Enrollment was performed by clinical investigators at participating centers, who had no role in the allocation process.”

4. Your comment 3b. Methods: There was only one stratification variable, age, used in the trial. Could the authors provide an assurance that there was good between-arm balance within the recruitment centres (12 tertiary-level hospitals) as centre was not a stratification variable?

Response: To address potential center effects, we conducted two post hoc analyses, i.e., Cochran-Mantel-Haenszel (CMH) test (stratified by site), logistic regression analysis (included the treatment group and site as covariates) to evaluate treatment effect on live birth. The CMH test showed that Breslow-Day test $P=0.997$, and adjusted $P = 0.035$. Logistic regression analysis showed that the P value for the site effect was 0.838, and P for the treatment was 0.035. The analyses were presented in **Supplement 2, Tables S8-S9**. Both analyses confirm consistent treatment effects across sites.

5. Your comment 3c. Methods: Sample size: the effect size/detectable difference is based on the observed difference in a previous study rather than a minimally clinically important difference, could the authors explain this? Also, the authors need to justify the choice of 80% power as opposed to 90%.

Response: In the field of reproductive medicine, a plausible and clinically important improvement is considered to be 5-10%. However, a single minimum clinically important difference (MCID) cannot be expected to be applicable in all cases. Our prior

RCT observed a 9% difference (42% vs. 33%), which aligned with the general guideline and provided an empirically derived effect size of our treatment ZYP, avoiding arbitrary MCID assumptions.

Both 80% and 90% power are reasonable choices. However, 80% appears to be a more common choice that balances the feasibility and statistical reliability. See, for examples, some recent trials on infertility (Hum Reprod. 2019;34(4):659-665.

doi:10.1093/humrep/dez017; Lancet. 2019;393(10178):1310-1318.

doi:10.1016/s0140-6736(18)32843-5; BMJ. 2025;388:e081474. doi:10.1136/bmj-2024-081474).

6. Your comment 4a. Statistical analysis: “Intention to treat” is ambiguous so authors need to clearly define the primary analysis set and the per-protocol set detailing who was excluded in the per-protocol set. Some information about this can be deduced from the results section in the CONSORT diagram, but this needs to be explicitly described.

Response: We employed an intention-to-treat (ITT) approach, which involves analyzing all randomized participants. Any transfer that didn't lead to a live birth, regardless of the reasons, a strategy commonly used in related trials (Lancet. 2019;393(10178):1310-1318. doi:10.1016/s0140-6736(18)32843-5; N Engl J Med. 2021;385(22):2047-2058. doi:10.1056/NEJMoa2103613). This conservative strategy fully incorporates the effects of ZYP across all ART stages, from ovarian stimulation and oocyte retrieval to fertilization, embryo transfer, implantation, and live birth, rather than isolating its impact on any single step.

The per protocol analysis excluded participants who had major protocol deviations, canceled a cycle or did not complete the pre-set minimum exposure dosage of the assigned study drug (at least 80% compliance). The definition has been added as a footnote of Table 4 for clarification.

7. Your comment 4b. Statistical analysis: A stratification variable was used in the randomisation. According to many published guidance documents stratification variables should usually be included as covariates in the primary analysis. The

presented analysis is unadjusted so this needs to be amended to reflect the recommended practice. Also, it is unclear how the confidence intervals were obtained for some of the univariable analyses reported such as the Chi-squared tests for the categorical variables.

Response:

With regards to stratification: Thank you for your insight. While we acknowledge the recommendation to include stratification variables as covariates in the primary analysis, our understanding is that the purpose of stratifying a variable during the randomization is precisely to eliminate (or at least minimize) its effect without considering it as a covariate while testing the primary hypothesis. However, randomization does not always work as perfectly as it is designed. In our trial, randomization achieved a good balance in baseline characteristics, including the stratification variable (age), minimizing confounding. While stratification during randomization improved allocation balance, we agree that it may not necessarily mandate covariate adjustments. To address the reviewer's concern, we conducted a sensitivity analysis adjusting for age using the Cochran-Mantel-Haenszel method, which yielded results consistent with the unadjusted analysis (adjusted $P = 0.040$ vs. unadjusted $P = 0.042$). Additionally, we performed prespecified subgroup analyses by age and comprehensively reported pregnancy outcomes to enhance transparency and facilitate clinical interpretation (Supplement 2, Table S1).

With regards to CI calculation: We apologized for the lack of clarity when calculating the relative ratio and ratio difference estimates. For relative risk (RR) estimates, the Delta method was used. For absolute difference in proportions, the Newcombe method was employed. We have implemented the above method in the Statistical Analysis section.

8. Your comment 4c. Statistical analysis: Handling of missing data: there are principled ways of dealing with missing data, but the authors seemed to have used methods such as best-case scenario replacing missing data with not having an event. Could the authors justify this and in addition perform some sensitivity analyses on

missing data.

Response: The primary outcome was defined as fresh cycle LBR. Our approach is actually most conservative and based on the worst-case scenario because we treated all missing outcomes as no live birth, giving rise to the minimal estimate of LBR. This is the exactly same strategy employed by many related studies (Lancet. 2019; 393(10178): 1310-1318. DOI:10.1016/s0140-6736(18)32843-5; BMJ. 2025; 388: e081474. DOI:10.1136/bmj-2024-081474). Please also see our response to your comment 5d.

9. Your comment 5a. Results: CONSORT diagram: a small typo- in the Figure 1. Study flowchart, in the Box 2, please change” 27 Not fulfilling the eligible criteria” to “27 Not fulfilling the eligibility criteria”.

Response: Thank you. We have corrected the typo and proofread the entire manuscript carefully.

10. Your comment 5b. Results: Baseline table: Some characteristics e.g. age, weight, height are almost identical between group which is not typical in real life RCTs. This is too good to be real, so could the authors verify that this is correct. Also, some of the reported standard deviation like SD for age seem to be quite small so the authors need to verify.

Response: We appreciate your concern. Participants were limited to: age 35-42 years, BMI<28 kg/m². These narrow ranges inherently reduced variability in baseline characteristics. So, what we reported was accurate and real. The two groups were indeed remarkably similar in these characteristics. We will make the raw data available to the public so that anyone can examine or use them.

11. Your comment 5c. Results: Duration of attempt to conceive shows a median difference (ZYP 3.0 (2.0-6.5), placebo 4.0 (2.0-7.0)) between groups of 1 year which is quite a long time. How was this variable recorded- in years or months?

Response: The duration of attempting to conceive was recorded in years, not months,

as reflected in the reported medians [ZYP: 3.0 (IQR 2.0 – 6.5) vs. placebo: 4.0 (IQR 2.0 – 7.0); $P = 0.119$, Wilcoxon test]. While the medians differ by one year, the means are similar (ZYP: 4.7 ± 3.9 vs. placebo: 4.9 ± 3.9), and the interquartile ranges show substantial overlap. These findings suggest that any imbalance is minimal and unlikely to have influenced the study outcomes.

12. Your comments 5d. Results: LBRs are quite different between the ITT and the per-protocol analyses- could the authors explain the implication of this?

Response: The primary outcome of this study was the live birth rate (LBR) following fresh embryo transfer. Participants who did not proceed to embryo transfer (ET)—due to cycle cancellation, treatment discontinuation, or other reasons—were naturally unable to achieve a live birth within the fresh cycle.

In the intention-to-treat (ITT) analysis, all randomized participants were included in the denominator ($n = 734$ vs. 733), irrespective of whether they underwent ET. In contrast, the per-protocol (PP) analysis excluded those who did not receive ET, resulting in a substantially smaller denominator ($n = 376$ vs. 374). This reduction of nearly half reflects the proportion of participants who either did not reach ET or did not meet the PP criteria.

Consequently, the ITT estimate of LBR is “diluted” by the inclusion of participants with no chance of achieving a live birth, leading to a lower overall rate. The PP analysis, however, focuses specifically on those who completed ET, thereby providing an LBR estimate among women who had a realistic opportunity of success in the fresh cycle.

13. Your comments 5e. Results: Could the authors relate the observed between group difference in the primary analysis and the assumed difference in the sample size calculation and discuss the implications in terms of study power?

Response: Our sample-size calculation assumed a 9% absolute difference per embryo transfer (ZYP 42% vs. placebo 33%) and targeted 80% power at $\alpha=0.05$. In the trial, the per-ET analysis showed a 6.8% absolute difference (40.5% vs. 33.7%; absolute difference 95% CI [0.2%~13.3%]), which is smaller than the assumed difference, hence

the achieved power for this effect size was below 80% (yet still sufficient to reach statistical significance). The ITT analysis yielded a 4.3% absolute difference (23.3% vs. 19.0%; absolute difference 95% CI [0.2~8.5]; P=0.042), reflecting dilution from participants who did not reach transfer, as expected for a fresh-cycle primary endpoint. Importantly, the observed effect size indicated that the true effect may be more modest than expected.

Reviewer #2:

1. Your comment 1: ZYP was provided 5 g once, three times daily around the time of embryo transfer. What is the rationale of such treatment protocol for the patients? There is not clear information provided regarding to the treatment period of ZYP or placebo.

Response: Sorry for the lack of clarity here. The dose was based on the instruction of Zishen Yutai Pill, and then was selected as the treatment dose. In our previous publications (Obstet Gynecol, 2022, 139, 192-201. DOI:

10.1097/AOG.0000000000004658), we used the same treatment protocol. We have also clarified the treatment period of ZYP or placebo in the Abstract. A more detailed description of treatment protocol can also be found in the Methods section. For your convenience of review, the description is enclosed here.

“Briefly, study drug intervention, ZYP or placebo, was administered 3 times daily, 5 g each time from down-regulation day (GnRH-a long protocol) or day 19 to day 23 of previous menstrual cycle (GnRH-ant protocol), until 2 weeks after ET. Study drug discontinued during the 1st day to the 4th day of menstrual cycle. Study drug intervention would continue when biochemical pregnancy was confirmed as positive. Intervention of study drug stopped in the case of negative pregnancy test. For patients achieved positive results in the pregnancy test, the study drugs were administered until the confirmation of clinical pregnancy (five weeks after ET). This treatment protocol was based on the instruction of the ZYP manufacturer and used in a previous study.”

2. Your comment 2: A total of 633 women (43.1%) did not receive embryo transfer or

dropped out of the study (312 in the ZYP group and 321 in the placebo group) before ET. The dropping rates are very high. Whether the dropping rates would bring impacts on data interpretation should be justified.

Response: We acknowledge that the high cycle cancellation rate was indeed a limitation of the study. We clarify below the reasons.

High cancellation rates are inherent in fresh-ET trials. The observed drop-out rate (~43%) is consistent with real-world ART practice, particularly in fresh-ET cycles. In AMA women, cancellations often occur due to diminished ovarian response (DOR) or failure to retrieve viable oocyte/embryos, while younger patients may cancel due to OHSS risk. Our previous large-scale trial (N=2,265) reported a similar cancellation rate (~42%), reinforcing that this is an expected phenomenon in such studies.

To account for the impact of high cancellation rates, we employed an ITT approach, analyzing all randomized participants regardless of whether they reached ET. This conservative strategy fully incorporates the effects of ZYP across all ART stages—from ovarian stimulation and egg retrieval to fertilization, embryo transfer, implantation, and live birth—rather than isolating its impact on any single step. By doing so, we provided a comprehensive assessment of ZYP’s “real” effectiveness, including scenarios where treatment discontinuation occurs.

ART trials, especially the fresh-cycle type, face unique statistical complexities due to the sequential milestones required for successful live birth (from oocyte retrieval, to fertilization, then ET, if fortunately, implantation, and hopefully live birth). ZYP could theoretically influence any of these stages (e.g., oocyte/embryo yield, fertilization, or miscarriage reduction), but ITT analysis avoids selective reporting bias by evaluating the overall outcome. Thus, while the dropping out rates would surely have impacts, our approach ensures that the results reflect clinical reality.

3. Your comment 3: Whether the ZYP has potential toxic effects on liver or renal functions? Safety issue is one of the concerns. Basic clinical biochemistry data should be provided.

Response: Our trial did not systematically assess liver and renal function through

biochemical testing, but we have the following safety information.

At first, in the current trial, adverse event monitoring revealed elevated ALT/AST in 2 participants from the ZYP group and 1 from the placebo group. One renal function-related adverse event was reported in ZYP group. Our prior study (involving 2,265 subjects) showed 1 case of elevated ALT/AST in the placebo group. No renal function abnormalities were observed in our previous trial.

Second, there was post-marketing surveillance of ZYP. An article identified one reported case of hepatic dysfunction associated with ZYP (Zishen Yutai Pill causing abnormal liver function: A case report. *China Journal of Modern Applied Pharmacy*, 2024; 41, 1818-1819. DOI: 10.13748/j.cnki.issn1007-7693.20231165). However, this case involved multiple concomitant medications, making it difficult to establish a direct causal relationship with ZYP alone.

Previous pre-clinical safety data showed that chronic toxicology studies in rats demonstrated no hepatotoxicity at doses up to 9.450 g/kg (6 times of the human equivalent dose) for 24 weeks in healthy rats and 16 weeks in CCl₄-induced liver injury models. Importantly, ZYP showed no aggravating effects on pre-existing liver injury (Xing et al., *Regulatory Toxicology and Pharmacology*, 2017, 83: 81-88. DOI: doi: 10.1016/j.yrtph.2016.12.001).

While these data suggest a favorable safety profile, we acknowledge that future studies incorporating routine liver and renal function monitoring would provide more comprehensive safety assessment. We thank the reviewer for highlighting this important consideration and will incorporate these measurements in our future research design.

4. Your comment 4: There is no previous marriage history and gravidity in the manuscript.

Response: Information on previous marriage history and gravidity has now been added to **Table 1**.

5. Your comment 5: With many herbal items in the ZYP, the authors should have

discussion about the potential interactions of different herbal items in promoting live birth.

Response: Several ZYP components have demonstrated benefits on ovarian function or endometrial receptivity in preclinical studies. *Panax ginseng* and *Lycium barbarum* have shown the ability to improve reproductive capacity in aged models (Frontiers in Endocrinol, 2022, 13:964069. DOI: 10.3389/fendo.2022.964069; Food & Function, 2024, 15(19):9779-9795: DOI: 10.1039/d4fo02720e). *Panax ginseng* improved follicular survival and maturation in mouse pre-antral follicle culture (Cell J, 2019, 21:100–106. DOI: 10.22074/cellj.2019.5733). *Rehmanniae Radix Praeparata* supported ovarian function through antioxidative and granulosa-cell effects (Frontiers in Pharmacol, 2024, 15:1426972. DOI: 10.3389/fphar.2024.1426972). *Eucommia Cortex* flavonoids modulated sex hormones and attenuated ovarian hyperplasia (J Ethnopharmacol. 2021;273:113947. DOI: 10.1016/j.jep.2021.113947). *Semen Cuscutae* flavonoids enhanced extravillous trophoblast invasion via Notch/AKT/MAPK signaling (Scientific Reports, 2018, 8:17342. DOI: 10.1038/s41598-018-35732-6). Of note, a previous study investigated the relationship between the chemical components of ZYP and their pharmacological efficacy using the endometrial receptivity disorder mouse model and premature ovarian failure mouse model. That study provided spectrum-effect relationship analysis, linking herb and chemicals to pharmacological activities. However, the precise interactions among the multiple herbal constituents and their combined effects on reproductive outcomes remain largely unresolved, highlighting the need for further mechanistic research.

A brief discussion has also been added in the Discussion section.

6. Your comment 6: Given that the papers published in Nature Communication are open accessed, cautious information should be provided to prevent the potential misuse of the information

Response: We appreciate the caution regarding the responsible use of open-access data. We have now included a statement in the discussion section, specifically under

“Limitations”. We have also called for appropriate clinical interpretation of the trial, emphasizing the considerations including the specific characteristics of our study population.

Reviewer #3:

Comment: I co-reviewed this manuscript with one of the reviewers who provided the listed reports. This is part of the Nature Communications initiative to facilitate training in peer review and to provide appropriate recognition for Early Career Researchers who co-review manuscripts.

Response: We appreciate the transparency regarding the co-review process. The comments from this co-reviewer were incorporated in the feedback of another reviewer.

Reviewer #4:

1. Your comment 1: The lack of proposed mechanism of action of the intervention or any features to support an area of mechanism e.g. difference in hormones or endometrial morphology makes it difficult to feel confident about the effects of the intervention. That said it was conducted in a double-blinded manner and a small benefit was found in LBR, although it remains unclear why/how this was achieved.

Response: In our study, we actually assessed endometrial morphology by measuring endometrial thickness on hCG trigger day, which showed no significant difference between groups (ZYP: 11.34 ± 2.50 mm vs. placebo: 11.17 ± 2.61 mm; $P = 0.258$). However, we recognize that endometrial thickness alone is an imperfect surrogate for endometrial receptivity, and a gold-standard clinical marker for receptivity remains elusive.

Beyond morphology, we observed trends toward improved embryo development in the ZYP group, including a higher number of available embryos (ZYP vs. placebo, median 4, IQR 2–5 vs. median 3, IQR 2–5; $P = 0.163$; [mean \pm SD: 4.0 ± 2.8 vs. 3.7 ± 2.6]) and high-quality embryos (ZYP vs. placebo, median 2, IQR 0–4 vs. median 2, IQR 0–4; $P = 0.178$; [mean \pm SD: 2.9 ± 3.1 vs. 2.6 ± 2.9]). Importantly, implantation rate was significantly increased in the ZYP arm (36.0% vs. 30.6%; $P = 0.022$), suggesting a

possible improvement in endometrial receptivity.

These clinical findings are supported by preclinical studies demonstrating that ZYP enhances endometrial receptivity by upregulating key implantation-related molecules such as HOXA10, ICAM-1, and SPP1, which play central roles in embryo adhesion and implantation (Evid Based Complement Alternat Med. 2015; 2015:317586; Heliyon. 2023; 9(9):e19395). Moreover, in a clinical study of patients with diminished ovarian reserve, ZYP improved embryo quality through modulation of oocyte-secreted factors BMP15 and GDF9 in the follicular fluid (Chin J Integr Med. 2023; 29(4):291–298). In addition to the pharmacological evidence supporting ZYP as a whole formulation, we have also discussed the efficacy of several individual herbs within ZYP on reproductive performance to provide more information on its potential mechanisms.

Collectively, these findings suggest that the observed improvement in live birth rate among AMA women treated with ZYP may be mediated by modest but cumulative benefits across both embryo quality and endometrial receptivity. While further mechanistic studies are needed, the convergence of clinical outcomes and preclinical evidence provides biological plausibility for ZYP's beneficial effect.

All these studies have been cited in the Introduction part, providing a brief introduction to the mechanical action of ZYP.

2. Your comment 2: Abstract - The first sentence does not set the scene well for the study. By 'inadequate' do you mean uncertain? As currently implies that it was previously investigated and found to not be effective.

Response: We elaborated our Abstract as follows, "Advanced maternal age (AMA, \geq 35 years) women undergoing assisted reproductive technology (ART) face reduced live birth rates (LBR) and remain a major clinical challenge. In a large randomized trial, Zishen Yutai Pill (ZYP) improved LBR in the general population, and a subsequent post hoc analysis suggested efficacy in AMA women, though it was underpowered to draw firm conclusions."

3. Your comment 3: Abstract - Population not clearly defined- AMA is not defined in

abstract. Other notable inclusion criteria should be presented.

Response: To address the comment about population definition, we have revised the abstract to explicitly define advanced maternal age (35-42 years) and BMI. Due to the strict 200-word limit for the abstract, we prioritized these clinically relevant criteria while omitting other details. The full manuscript provides additional details about the study population.

4. Your comment 4: Abstract - 'around the time of embryo transfer' is vague.

Response: We have stated the exact time of intervention in the abstract as follows, "Women aged 35-42 years with BMI <28 kg/m² at 12 tertiary-level hospitals in China were randomly assigned to receive ZYP or placebo orally (5 g once, three times daily) from day 19-23 of the preceding menstrual cycle until 2 weeks after embryo transfer, continuing if biochemical pregnancy was confirmed."

5. Your comment 5: Introduction - "Nevertheless, advanced age is conclusively associated with a reduced live birth rate (LBR)" Important to explain the reason for this statement- ie poorer oocyte quality as opposed to reduced implantation, as your intervention seems to be aimed at the latter given that it is applied 'around the time of embryo transfer'? You have not shown any differences in oocyte / embryo quality nor in endometrial measures?

Response: Please see our response to your Comment 1 above.

6. Your comment 6: Introduction - "Nevertheless, AMA women had less favorable reproductive tract conditions, altered hormone levels and poorer ovarian reserve compared to the non-AMA women." – Please specify more clearly what you are referring to?

Response: In this sentence, we intended to summarize the multifaceted reproductive challenges commonly observed in AMA women of advanced maternal age, including structural, hormonal, and ovarian factors.

At first, by "poorer ovarian reserve", we intended to reflect the age-related decline in

both the quantity and quality of oocytes, characterized by reduced antral follicle counts, lower anti-mullerian hormone (AMH) levels. This decline contributes to poorer response to ovarian stimulation, and an increased risk of aneuploid embryos, ultimately reducing the likelihood of a successful live birth (J Ovarian Res, 2023, 16: 67. DOI: 10.1186/s13048-023-01151-z).

By "less favorable reproductive tract conditions," we refer to the increased prevalence of uterine structural abnormalities such as fibroids, adenomyosis, and endometrial polyps, as well as a higher incidence of prior cesarean sections and pelvic inflammatory disease, all of which can compromise uterine anatomy and function (Facts Views Vis Obgyn, 2020, 12(2): 91-104 [DOI not available]). Additionally, AMA is associated with reduced endometrial receptivity, partly due to altered gene expression and impaired decidualization, which may hinder embryo implantation and increase the risk of complications such as ectopic pregnancy and placental abnormalities (Hum Reprod Update. 2023, 29, 773-793. DOI:10.1093/humupd/dmad019).

By "altered hormone levels," we mean that AMA women often exhibit disrupted endocrine profiles, including declining progesterone levels, attenuated estrogen and progesterone receptor expression in the endometrium, all of which can negatively impact implantation and early pregnancy support (Hum Reprod Update. 2023, 29, 773-793. DOI:10.1093/humupd/dmad019).

Together, these interrelated factors highlight the complex and multifactorial nature of age-related reproductive decline in AMA women, providing a biological basis for the observed differences in reproductive outcomes compared to younger women.

The sentence is elaborated in the Introduction, with more references cited.

7. Your comment 7: Introduction - How does Zishen Yutai Pill work? What is the biological or scientific basis for the usage of this supplement in women of advanced maternal age?

Response: Please see our response to your Comment 1 above. We have also added a brief discussion regarding the constituents of ZYP, providing more evidence on biological and scientific basis for the use of ZYP in AMA women.

8. Your comment 8: Introduction - “The post hoc subgroup analysis from our previous trial provided promising evidence for the effect of ZYP on AMA women in ART.” Please describe what data this refers to in more detail (samples size and effect size in what parameter) as most readers will not be familiar with your previous trial results. Add reference to your previous trial here also.

Response: The post hoc subgroup analysis from our previous trial provided evidence for the effect of ZYP on AMA women in ART (ZYP vs. placebo: clinical pregnancy rate, 33.0% [68/206] vs. 23.1% [51/221], absolute difference and 95% CI: 9.9% [1.4%, 18.3%], P = 0.022; LBR, 26.2% [54/206] vs. 19.9% [44/221], absolute difference and 95% CI: 6.3% [-1.7%, 14.3%], P = 0.122). We have cited our previous research, and also included the samples size, the pregnancy outcomes and their effect sizes in the Introduction section, for better understanding of the readers.

9. Your comment 9: Methods - Should read ‘The protocol is available’

Response: We have revised the text as suggested.

10. Your comment 10: Methods - ‘both ovaries existed’ should read are present.

Response: We have revised the text as suggested.

11. Your comment 11: Methods - ‘by following the Human Fertilisation and Embryology Authority.22’ should read ‘in line with categories described by the HFEA’ or similar.

Response: We have revised the text as suggested.

12. Your comments: Methods - Exclusion criteria of: “the uterine cavity line constricted by uterine fibroids”. This is not very clear – does this represent submucosal fibroids, distortion of the cavity by an intramural fibroid, or something else?

Response: In the original protocol, “the uterine cavity line constricted by uterine fibroids” referred to submucosal fibroids or intramural fibroids that distorted the endometrial cavity, as identified by transvaginal ultrasound or hysteroscopy.

13. Your comment 13: Methods - Likewise, in the exclusion criteria: “had uncured endometrial diseases” should be defined more clearly.

Response: “Uncured endometrial diseases” referred to active or unresolved intrauterine pathologies that could adversely affect implantation, including endometrial polyps, chronic endometritis, intrauterine adhesions, and endometrial hyperplasia where treatment had not yet been completed or confirmed effective. We have revised the manuscript text to make these definitions explicit for clarity.

14. Your comment 14: Methods - Describe the protocol in more detail. What formulations of FSH were used etc. How was the choice of suppression protocol decided and why was it not standardised for all patients? In turn, would this impact the total dose of ZYP administered?

Response:

With regards to FSH: FSH was used, including urine-derived follicle-stimulating hormone (uFSH), and recombinant human follicle-stimulating hormone (rFSH). In addition, urine-derived human menopausal gonadotropin (HMG) (containing both FSH and LH) was also used.

For your convenience, we also report the distribution of FSH-related reagents used between groups in ITT population: uFSH (51.8% vs 49.7%, $P = 0.418$), and rFSH (60.8% vs 56.8%, $P = 0.119$). The distribution of HMG usage was 27.5% in ZYP vs. 26.6% in placebo ($P = 0.693$).

With regards to suppression protocol and the total dose of ZYP:

The GnRH-a long protocol and GnRH antagonist protocols are the two most commonly used protocols for women of advanced age in China (A Chinese practice guideline of the assisted reproductive technology strategies for women with advanced age, *J Evid Based Med.* 2019, 12:167-184. DOI: 10.1111/jebm.12346), and were also the main protocols adopted by participating centers. Allowing both protocols reflects real-world clinical practice, and thus improved the feasibility of patient recruitment across centers,

and enhanced the generalizability of our findings.

The total dose of ZYP administered was not affected by the choice of suppression protocol. Study drug intervention was administered 3 times daily, 5 g each time from down-regulation day (GnRH-a long protocol) or day 19 to day 23 of previous menstrual cycle (GnRH-ant protocol), until 2 weeks after ET. Study drug discontinued during the 1st day to the 4th day of menstrual cycle. Study drug intervention would continue when biochemical pregnancy was confirmed as positive. Intervention of study drug stopped in the case of negative pregnancy test. For patients achieved positive results in the pregnancy test, the study drugs were administered until the confirmation of clinical pregnancy (five weeks after ET).

The dosing schedule and duration were aligned between protocols, ensuring comparable total exposure.

15. Your comment 15: Methods - Why was 28 weeks chosen as the cutoff for live birth outcome as opposed to 24 weeks, which is generally considered to be the cutoff of viability? Please use definitions in line with the CORE outcome set. [https://www.fertstert.org/article/S0015-0282\(20\)32680-7/fulltext](https://www.fertstert.org/article/S0015-0282(20)32680-7/fulltext)

Response: We are aware of and appreciate your reference to the other definition. Our definition of live birth follows the Chinese textbook of Obstetrics and Gynecology, as “Live birth was defined as delivering any viable infant at 28 weeks or more of gestation”. (Obstetrics and Gynecology, People’s Medical Publishing House, 2018, ISBN978-7-117-26439-6, p95). And this definition was a common one used by Chinese practitioners. Other researchers in China had also used same definition in their work (N Engl J Med, 2016, 375(6):523-533; doi: 10.1056/NEJMoa1513873). For our trial, there were no live births from 24 to 28 weeks. Therefore, the conclusion is the same if we used the 24-week as the definition of live birth.

16. Your comment 16: Methods - Clarify that this was live birth per cycle started?

Response: We have clarified the definition of the primary outcome as: “Live birth was defined as the delivery of any viable infant after 28 weeks of gestation in a fresh embryo

transfer cycle”. However, it may not be entirely accurate to describe this outcome as “live birth per cycle started”. In our study, the live birth rate was calculated based on the ITT population, which includes all participants as randomized. The ITT-based live birth rate differs conceptually from “per cycle started”, as the latter one refers specifically to the number of cycles initiated, rather than the number of participants randomized.

17. Your comment 17: Methods - “Clinical pregnancy was defined as the presence of intrauterine gestation sac with fetal motion under transvaginal ultrasonography” - does this refer to detectable fetal heart activity?

Response: Yes. This has been clarified in the Methods section on Outcomes.

18. Your comment 18: Method - Under what circumstances was a combination of IVF+ICSI used for fertilization?

Response: In our study, a combination of IVF and ICSI was used for patients at risk of low or no fertilization with conventional IVF to mitigate this risk. This included women with unexplained infertility for 7 years or longer, as well as those whose husbands had a history of significant fluctuations in sperm concentration or motility. The combined approach was intended to maximize the likelihood of successful fertilization in these higher-risk scenarios.

19. Your comment 19: Method - Why was ET performed from D2 – D4 ie non-blastocyst transfer for the majority rather than day 5 blastocyst transfer?

Response: During the study period, cleavage-stage embryo transfer was more commonly performed in AMA women in China. In line with Chinese ART clinical practice, transferring two cleavage-stage embryos was a common strategy aimed at balancing the chance of achieving pregnancy with the need to control multiple pregnancy rates.

20. Your comment 20: Method - Can you specify the exact conditions of the Per-

Protocol analysis? This is unclear also in Table 4.

Response: We have stated the conditions of the per-protocol analysis in the footnote of Table 4 as follows: “Per-protocol (PP) analysis was performed, excluding those who had major protocol deviation(s), canceled cycle or did not complete the pre-set minimum exposure dosage of the assigned study drug (at least 80% compliance), from the ITT population.”

21. Your comment 21: Figure 1 should also describe the suppression protocol split.

Response: We have added the suppression protocol split in Fig. 1.

22. Your comment 22: It is still not clear how the investigative medicinal product being investigated is supposed to work? Any useful background information would be useful to describe.

Response: As mentioned in our response to your Comment 1 above, background information of ZYP has been added in the Introduction section.

23. Your comment 23: ‘controlled ovarian hyperstimulation (COH)’ is an outdated term and can now just be described as ovarian stimulation.

Response: We have corrected the term as you suggested.

24. Your comment 24: Methods - Power- “In our previous clinical research, the LBR was 0.42 per embryo transfer in ZYP treated group and 0.33 per embryo transfer in placebo control group among AMA women.¹⁹” State the sample size this statistic is based on. Usually, the power is derived to the ‘minimum clinically important difference’ (MCID) rather than to the expected change in LBR with a particular intervention.

Response: The sample size calculation was based on our prior RCT, which demonstrated a clinically relevant 9% increase in LBR (42% vs. 33%) among patients with embryo transfer—consistent with reproductive medicine’s MCID threshold (5–10%). This approach avoids arbitrary MCID assumptions by using empirically derived effect sizes while maintaining methodological credibility. The actual and observed LBR

among patients with embryo transfer turned out to be 6.8% which is lower than we assumed but still consistent with the reproductive medicine's MCID range of 5 to 10%.

25. Your comment 25: Results - The reason for not having embryo transfer should be explained more clearly e.g. what is meant by 'Abnormal endometrium', and specify level for 'Elevated E2/P'. What does 'Unwilling to participate' mean - as in withdrew consent?

Response: In our study, "abnormal endometrium" was defined as the presence of any of the following conditions prior to embryo transfer: thin endometrium (<7 mm), uterine cavity effusion, or abnormal uterine bleeding. "Elevated E2/P" was defined as serum estradiol (E2) levels ≥ 3000 pg/ml and/or serum progesterone (P) levels ≥ 1.5 ng/ml. "Unwilling to participate" meant withdrew consent. These have been clarified in Figure 1.

26. Your comment 26: Results - For completeness, please present comprehensive metrics of the variables in Tables 1 and 2: Min, Mean, SD, Median, Q1, Q3, IQR, Max, N.

Response: We have provided a comprehensive metrics of variables in Tables 1 and 2, and in the **Supporting Information 2, Tables S6-7**.

27. Your comment 27: Results - Table 2: is Gonadotrophin dose referring to the total dose during OS?

Response: Yes. This has been clarified in the footnote of Table 2.

28. Your comment 28: Results - Were patients taking oral ZYP in previous IVF attempts (and how long ago)?

Response: No. According to the inclusion criteria, we excluded participants who had TCM therapy (including ZYP) for infertility within one month.

29. Your comment 29: Results - Table 2: missing "s" on Number of available embryos.

Response: We have revised this as you suggested.

30. Your comment 30: Results - Table 2: the outcome of mature oocytes retrieved is assumingly only relevant to the ICSI cohort? Please specify this as a footnote indicating number of relevant cycles. This is difficult to understand with the "IVF+ICSI" cohort especially.

Response: According to our protocol, for ICSI cycles, the number of mature (MII) oocytes was recorded on the day of retrieval, whereas for conventional IVF cycles, oocytes showing pronuclei or a second polar body on Day 1 after retrieval were retrospectively considered mature. The definitions are available in the protocol.

At the time of study design, we regarded the IVF Day-1 pronuclear/second polar body count as an indication of oocyte maturity (see Supplement 1, pp 16). However, we acknowledge that the reviewer's definition of mature oocytes aligns with current consensus. Our original approach may have deviated from this standard. Therefore, we have removed this row of data to avoid potential misinterpretation. We have added this as a limitation of our trial.

31. Your comment 31: Results - Differences in pregnancy loss are insignificant between the two groups which does not align with ZYP being approved for treatment of miscarriage (ref 26)?

Response: First, we wish to note that in reproductive medicine, interventions generally yield small effect sizes in outcomes such as live birth or miscarriage prevention, requiring large sample sizes to detect statistically significant differences. For example, in the well-powered PRISM trial involving 4,153 women, progesterone treatment resulted in a modest increase in live birth rate (75% vs. 72%; relative rate 1.03; 95% CI, 1.00 - 1.07; P = 0.08). In our trial, among those who achieved biochemical pregnancy (ZYP: 232; Placebo: 195) and clinical pregnancy (ZYP: 216; Placebo: 177), the sample size was substantially smaller and therefore underpowered to detect potential differences in miscarriage rates.

Second, the population enrolled in our study differs from that typically included in

miscarriage prevention trials. Specifically, our participants were not selected based on vaginal bleeding during early pregnancy.

Thus, the insignificant difference in pregnancy loss in our study does not contradict ZYP's previously reported benefits in miscarriage treatment, but rather reflects both a limitation in statistical power and a difference in the target population.

32. Your comment 32: Results - Did the live birth rate results remain the same when analysis was performed depending on the type of stimulation cycle or stage of embryo transfer Ie Cleavage stage only or Blastocyst transfer only?

Response: Subgroup analyses were performed among participants with embryo transfer according to the type of stimulation cycle or the stage of embryo transfer, the treatment effect remained consistent across subgroups (Breslow–Day test for homogeneity: P = 0.602 for stimulation cycle type, P = 0.738 for embryo transfer stage). After statistical adjustment, the P values for the interaction remained significant (adjusted P = 0.046 and P = 0.048, respectively).

Outcome	ZYP group (n=422)	Placebo group (n=412)	P value
Live birth	171 (40.5)	139 (33.7)	0.043
Type of stimulation cycle			
Long protocol	139/342 (40.6)	111/319 (34.8)	0.121
Antagonist protocol	32/80 (40.0)	28/93 (30.1)	0.173
Stage of embryo transfer			
Cleavage stage	125/326 (38.3)	102/326 (31.3)	0.059
Blastocyst	46/96 (47.9)	37/86 (43.0)	0.508

Data are presented as n (%).

33. Your comment 33: Was the type and dose of luteal phase support administered standardized amongst all patients who received an embryo transfer?

Response: The type and dose of luteal phase support were not strictly standardized but were comparable between the two groups. As shown in the table below, the proportions of patients receiving oral, intramuscular, or vaginal progesterone were comparable in

ZYP and placebo group, with no statistically significant differences in administration route or dosage. Median doses and interquartile ranges were also consistent across groups for each administration route, indicating balanced luteal phase support regimens between the study arms.

Table. Route and dose of luteal support in the study

Route and dose of luteal support	ZYP Group (N=422)	Placebo Group (N=412)	P
Participants receiving oral support	373 (88.4)	376 (91.3)	0.170
Dose of oral support/mg	2940 (335-7740)	3400 (340-7720)	0.946
Participants receiving intramuscular support	19 (4.5)	12 (2.9)	0.225
Dose of intramuscular support/mg	840 (320-1120)	900 (360-1120)	0.921
Participants receiving vaginal support	190 (45.0)	187 (45.4)	0.916
Dose of vaginal support/mg	1530 (1350-1530)	1530 (1350-1530)	0.800

Data are presented as n (%) or median (IQR range).

34. Your comment 34: Did you observe any differences in oocyte quality or embryo quality in this RCT?

Response: We observed trends, but not statistically significant, toward improved embryo development in the ZYP group compared with placebo, including a higher number of available embryos (median 4, IQR 2–5 vs. median 3, IQR 2–5; P = 0.163; mean ± SD: ZYP 4.0 ± 2.8 vs. 3.7 ± 2.6) and high-quality embryos (median 2, IQR 0–4 vs. median 2, IQR 0–4; P = 0.178; mean ± SD: ZYP 2.9 ± 3.1 vs. 2.6 ± 2.9).

35. Your comment 35: Discussion - “In the field of ART, even modest improvements in LBR can have profound clinical significance, with a 5–10% increase widely recognized as the minimal clinically important difference”. This argues against the power calculation that you used for this study as much larger numbers would be needed to detect an MCID of 5%.

Response: We acknowledge that in the field of reproductive medicine, a 5–10% increase in LBR is often regarded as the minimal clinically important difference (MCID). However, a single MCID threshold cannot be universally applied across all research contexts. In our prior RCT, we observed a 9% absolute difference in LBR (42%

vs. 33%) with ZYP treatment among patients with embryo transfer, which falls within this MCID range. We therefore based our power calculation on this empirically derived effect size, rather than an arbitrarily assumed MCID. The actual and observed LBR among patients with embryo transfer turned out to be 6.8% which is lower than we assumed but still consistent with the reproductive medicine's MCID range of 5 to 10%. To be conservative and follow the common practice, we used the ITT strategy in testing the primary hypothesis, and the ITT analysis still detected a difference of 4.3% with a p-value of 0.042. In our case, if we enrolled many more patients, the result might be more statistically significant, but the conclusion would be similar as can be seen in **Supplement 2, Table S3** (pooling data from our two randomized clinical trials, ZYP: 940 vs. Placebo: 954, fresh cycle LBR: 23.9% vs. 19.2%, P = 0.012, absolute difference and 95% CI: 4.8 [1.1~8.5]).

36. Your comment 36: Discussion - "In addition, both the per-protocol analysis (41.0% vs. 33.7%, P=0.040) and analysis among women with embryo transfer (40.5% vs. 33.7%, P=0.043) revealed a 7% absolute increase in LBR." How can this be true? Per protocol means including everyone who started the protocol as the denominator ie from 734, so it will be much lower than when using only those who had embryo transfer eg 432.

Response: We apologize for the confusion with the denominators. The ITT population comprises all randomized participants (ZYP 734 vs Placebo 733); the PP population excludes major protocol deviations, cancelled cycles, and participants with <80% compliance (PP N = 376 vs 374); and the participants with embryo transfer analysis uses those who actually reached embryo transfer (N = 422 vs 412). Because the PP and embryo-transfer denominators are similar in magnitude, the absolute LBR differences computed on those denominators are also similar (PP: 41.0% vs 33.7%; embryo-transfer: 40.5% vs 33.7%), both yielding ~7% absolute increases. We have clarified these definitions and the denominators of ITT and PP in Statistical analysis and Table 4 to avoid this misunderstanding.

37. Your comment 37: Discussion - “IVF vs. ICSI in non-severe male factor infertility (4.1% absolute difference in cumulative LBR favoring IVF.34, 35, 36” The difference in LBR in one cycle is not comparable to a cumulative rate.

Response: We agree that a one-cycle live birth rate is not directly comparable to a cumulative live birth rate. Our intention was to illustrate the magnitude of effect in the context of other well-known findings in reproductive medicine. We have revised this section to ensure that all comparisons are restricted in LBR in one cycle, and have removed references regarding cumulative LBR to avoid potential misinterpretation.

38. Your comment 38: Quality of language can be improved throughout, particularly with errors in usage of tense in all sections apart from the results/discussion.

Response: We have proofread our manuscript in its entirety carefully to improve the quality of language.

39. Your comment 39: Abstract- grammatical issues eg: “Although Zishen Yutai Pill (ZYP) have been demonstrated benefits to women undergoing ART” “...its efficacy on the live births of advanced maternal age women remained inadequate”

Response: The Abstract section has been rewritten and the entire manuscript has been carefully proofread.

40. Your comment 40: Intro – issues with tense: “Advanced maternal age (AMA) is defined as the maternal age of 35 years or older” “In recent years, the proportion of AMA women undergoing autologous ART had increased from 63.0% in 2014 to 71.8% in 2019 around the globe.” Page 6 “ZYP had recently been reported to improve the LBR among women undergoing IVF/ICSI.”

Response: We have maintained the present tense for definitions. For example, “Advanced maternal age (AMA) is defined as the maternal age of 35 years or older”. For factual reporting, we have decided to use simple past tense. For example, “In recent years, the proportion of AMA women undergoing autologous ART increased from 63.0% in 2014 to 71.8% in 2019 around the globe.” “ZYP was recently reported to improve

the LBR among women undergoing IVF/ICSI.”

41. Your comment 41: Participants: “both ovaries existed”

Response: We have corrected as you suggested.

Reviewer #5:

1. Your comment 1: Check the tense as the abstract writes like it’s a protocol “aims to evaluate

Response: We have revised the tense throughout the abstract and manuscript.

2. Your comment 2: “(5 g once, three times daily)” do you mean 5g three times daily?

Response: The dosage was actually 5 g per dose, three times daily; a total of 15 g/day.

3. Your comment 3: Abstract should state brief def of AMA (>35)

Response: We have added a brief definition of AMA in the abstract as “advanced maternal age (AMA, age \geq 35 years)” to ensure clarity.

4. Your comment 4: Please add a description of ZYP to the background including its proposed therapeutic effect. Is it proposed to improve stimulation response? Embryo quality? Endometrial quality? Etc There is some of this in the Discussion, but readers need it in the intro to understand the study design and question

Response: We have added a brief background in the Introduction section. In addition to the pharmacological evidence supporting ZYP as a whole, we have also discussed the efficacy of several individual herbs within ZYP on reproductive performance to provide more information on its potential mechanisms.

5. Your comment 5: I’m not sure its entirely true that “their effects on pregnancy outcomes, particularly live births, remained unexplored” you will find each of these interventions is covered by a corresponding Cochrane review and that clinical outcomes

including pregnancy and live birth are reported in many RCTs – these outcomes are not entirely unexplored.

Response: We have revised the sentence to clarify that our statement refers specifically to the lack of evidence for live birth outcomes in AMA women. In the Cochrane review, none of the included studies assessed live birth in AMA women, although many RCTs in non-AMA populations have reported pregnancy and live birth outcomes.

6. Your comment 6: The study was driven by post hoc analysis of an RCT by the same team which showed possible benefit of ZYP for AMA. That study recruited 2014-2017 and was published in 2019. This study recruited from March 4, 2019 to October 14, 2023. However the study appears to have been funded in 2017 based on the funding references provided (20174002 and 201704020046) and the funding agreement provided in the supp files “Project start and end time: January 2017 to December 2019”– please explain.

Response: In our previous trial, recruitment was performed between April 2014 and June 2017, with follow-up completed in June 2018 (Obstet Gynecol, 2022, 139, 192-201. DOI: 10.1097/AOG.0000000000004658). This earlier study provided the most direct basis for the present trial, including experience in multicenter coordination, the trial protocol framework, and an empirically derived effect size for ZYP in improving pregnancy outcomes among AMA women.

However, the concept for this study was formed much earlier. Since 2012, ZYP had been used in small-scale clinical practice in our leading center, during which we observed a potential benefit for pregnancy outcomes, particularly in AMA women. These preliminary observations prompted us to apply for the two grants mentioned in the manuscript (20174002: January 2017–December 2019; 201704020046: May 2017–April 2020). It is important to note that the dates stated here represent the initially planned execution periods for the projects, not the actual completion dates of this trial. It turned out that both funding projects underwent extensions beyond their original timelines. This happens quite often not only in China but also in the United States.

After receiving funding, we began an extended preparation phase, which included multicenter site surveys, coordination meetings, and detailed trial design. By the first half of 2017, recruitment for our earlier RCT had already been completed. Live birth outcome data were fully collected in June 2018. Then, our previous trial was unblinded. We waited for the statistical analysis of that RCT to confirm ZYP's effect in AMA women before launching the present trial. During this period, we continued the preparatory work for the multicenter study, including site coordination, preparation of ethics submissions and study drugs, and protocol refinements, which required considerable time before the current trial commenced in 2019.

7. Your comment 7: Another possible inconsistency is that recruitment commenced March 2019 but the protocol provided in supp files was written in May 2019. Please explain.

Response: We apologize for the confusion resulting from the trial recruitment start date (March 2019) and the protocol approval date (May 2019). Our trial began the enrollment following the approval of an earlier version of the protocol, which was updated, finalized, and approved in May 2019 to incorporate minor but necessary changes based on peer feedback and practical considerations after early implementation. The key updates included two parts.

1. Primary outcome revision. Originally, the protocol included both clinical pregnancy rate and live birth rate as co-primary outcomes. In the final version, the primary outcome was set as live birth rate alone, as it is the most clinically relevant endpoint in ART trials. Notably, the original sample size calculation was solely based on live birth rate.

2. Clarification on embryo quality criteria. During early multicenter recruitment, variability in embryo grading standards across sites was observed. The final protocol added explicit definitions to high-quality embryos.

However, inclusion/exclusion criteria, randomization procedures, and blinding methods remained identical to the initial version.

8. Your comment 8: The intervention and placebo were manufactured and provided for free by Guangzhou Baiyunshan Zhongyi Pharmaceutical Co. Ltd and this company also hired Guangzhou Evidence-Based Medicine Tech Co. Ltd to do the data collection and randomisation etc. What role did Guangzhou Baiyunshan Zhongyi Pharmaceutical Co. Ltd have in the study protocol and monitoring then? Did the instruct the EBM Tech co in the protocol etc? Their involvement should be clear as they have an obvious COI as do the investigators who receive grant funding from them

Response: Guangzhou Baiyunshan Zhongyi Pharmaceutical Co. Ltd. provided the study drugs and funded the independent CRO (Guangzhou Evidence-Based Medicine Tech Co. Ltd.) for trial execution. Importantly, the CRO independently handled all randomization processes, including drug numbering, packaging, and dispensing. Neither the investigators nor the sponsor had access to randomization codes or treatment allocation, preserving blinding of participants, clinicians throughout the trial. The sponsor's involvement was restricted to quality assurance (e.g., providing drug-related materials to investigators and monitoring enrollment progress), with no involvement in the protocol design, patient consent and enrollment, data analysis, or manuscript preparation. Funding sources (including grants from Guangzhou Baiyunshan Zhongyi and government programs) were transparently disclosed, while scientific independence was preserved by investigator-led decision-making. We have further clarified these in "Randomization and Masking" in the Methods section, and the "Funding" and "Competing Interests" sections.

9. Your comment 9: There are 6 pregnancies resulting in cycle cancellation in each arm per the flow chart – are these included in the total numbers pregnant? Please make it clear.

Response: These pregnancies were excluded in the total number of pregnancies. The live birth rate was actually fresh cycle LBR. This definition is available in the Abstract, as well as the Method section. Thus, those pregnancies due to natural conception and subsequent frozen-thawed embryo transfer were all not included in the fresh-cycle ITT analysis.

10. Your comment 10: Methods section states “The trial protocol received approval from the ethics committees of all sites” please give the name and reference number for the main ethics approval. I see this is provided elsewhere but it should be clear in the methods

Response: We have provided detailed information for the ethics committees of all sites in Supplement 6.

11. Your comment 11: Methods – Participants “with modifications in age and body mass index (BMI).” What does this mean? You modified the inclusion criteria during the trial? Or there were modifications compared to your other studies?

Response: The phrase 'with modifications in age and body mass index (BMI)' refers to differences between the inclusion/exclusion criteria used in this trial compared to those in our previous related studies (Obstet Gynecol, 2022, 139, 192-201. DOI: 10.1097/AOG.0000000000004658)—not modifications during the current trial. The criteria were predefined before trial initiation and applied consistently. We have clarified the wording as follows: “The following inclusion and exclusion criteria were based on our previous research, but with differences in age and body mass index (BMI) compared to our previous studies.”

12. Your comment 12: “permuted block randomization method.” – what were the block sizes? Please state.

Response: The size of block was 4. We have provided this information in the Methods section of the manuscript.

13. Your comment 13: When was randomisation carried out? What day in relation to the IVF cycle?

Response: Randomization was performed prior to the start of the ovarian stimulation cycle (see Supplement 1. Protocol, pp. 12). We have clarified this in the Methods section of the manuscript.

14. Your comment 14: The sample size is a little unclear – please provide the expected LBR in each arm after applying the dropout rates etc.

Response: The expected live birth rates (LBR) of 42% in the ZYP group and 33% in the placebo group were assumed per embryo transfer based on our previous study. To achieve 80% power with a two-sided significance level of 0.05 to detect this difference in LBR per embryo transfer, we estimated that 454 participants per arm would be required who actually underwent embryo transfer.

Considering the dropout rate (15%) and cycle cancellation rate (23%) before embryo transfer in AMA women, we anticipated that approximately 38% of enrolled participants would not reach embryo transfer. To account for the expected 38% attrition (dropout and cancellation), we inflated the sample size to at least 733 participants per arm. Please see the description in the Method section.

15. Your comment 15: Per protocol analysis excluded people with cancelled cycles. Is this appropriate? If the ZYP was administered from day 19 it could conceivably impact on the chance of cancellation. Indeed the investigators observed more cancellations in the placebo group. Cancellations for personal reasons may be ok to exclude but I would have expected to include cancellations for poor response/no embryos etc.?

Response: The primary outcome of this study was live birth in fresh embryo transfer (ET) cycles, so participants who did not reach ET inherently could not contribute to this endpoint. Therefore, excluding canceled cycles from the PP analysis is appropriate and consistent with evaluating treatment efficacy among those eligible to achieve the primary outcome.

While ITT analysis includes all randomized participants and captures the overall clinical effectiveness including cancellations, the PP analysis focuses on those who completed the protocol sufficiently to evaluate the direct effect of ZYP on live birth after fresh ET. This same strategy has been used in related studies (N Engl J Med. 2016; 375(6): 523-533. doi:10.1056/NEJMoa1513873; BMJ. 2025; 388: e081474. doi:10.1136/bmj-2024-081474).

Additionally, medication compliance rate $\geq 80\%$ is a common standard in the field of TCM research (JAMA. 2023; 330(16): 1534-1545. doi:10.1001/jama.2023.19524; JAMA Intern Med. 2024; 184(7): 727-735. doi:10.1001/jamainternmed.2024.1190).

16. Your comment 16: There were more multiple embryo transfers in the ZYP group. Why? Could this have contributed to the apparent benefit of ZYP? If appropriate it might be helpful to include the mean number of available and high-quality embryos so we can see if there is any difference, as if not, there is no explanation for more multiple transfer in the ZYP group?

Response: Thank you for raising this important point regarding the higher rate of multiple embryo transfers in the ZYP group. To clarify terminology, we distinguish between “multiple embryo transfers” (referring to multiple transfer cycles per patient) and “transfers of multiple embryos” (meaning two or more embryos transferred in a single cycle). Since the primary endpoint was fresh cycle LBR, our observation relates to the latter—the ZYP group had more cycles in which two or more embryos were transferred simultaneously compared to placebo.

This difference appears to be related to trends toward improved embryo development with ZYP treatment. Specifically, the ZYP group showed a higher median number of available embryos (4 [IQR 2–5]) versus placebo (3 [IQR 2–5]), although this difference did not reach statistical significance ($P = 0.163$). The mean number of available embryos also tended to be greater in the ZYP group (4.0 ± 2.8 vs. 3.7 ± 2.6). Similarly, the number of high-quality embryos was slightly higher in the ZYP group (median 2 [IQR 0–4], mean 2.9 ± 3.1) compared to placebo (median 2 [IQR 0–4], mean 2.6 ± 2.9), with a non-significant trend ($P = 0.178$).

Most importantly, the proportion of participants who had at least two available embryos at the time of embryo transfer was significantly higher in the ZYP group (399/422, 94.5%) than in placebo (371/412, 90.0%, $P = 0.015$). This likely contributed to the increased rate of multiple embryo transfers observed in the ZYP group, supporting the hypothesis that improved embryo availability may partially explain the treatment benefit.

17. Your comment 17: Discussion states “In the current research, we pooled data from two clinical trials to validate the efficacy of ZYP on fresh cycle live births among AMA women. The results were robust among the ITT population (Table 3), PP population (Table 4)...” this implies that Table 3 and 4 contain the data from two trials pooled – which I don’t understand to be the case. Please rephrase.

Response: In the current research, we aimed to validate the efficacy of ZYP on fresh cycle live births among AMA women. **Tables 3** and **4** (ITT and PP populations, respectively) and **Table S2** (population with embryo transfer) present the original results from the current trial only, without pooling data from the other study. In addition to this primary analysis, we conducted a separate pooled analysis that combined data from both our current trial and our previous trial to further assess the robustness and generalizability of the findings. The pooled analysis results, presented in **Supplement 2, Tables S3** and **S4**, were consistent with the current trial results, showing comparable effect sizes. We have rephrased the discussion for clarification. We can also consider removing this discussion if the reviewer feels it would be better.

18. Your comment 18: Discussion states “The ZYP was approved for the treatment of threatened miscarriage and recurrent pregnancy loss” which implies its some sort of regulatory or guideline approval, but the reference is to a meta-analysis. Please check and consider rephrasing.

Response: ZYP has indeed been approved for treatment of threatened miscarriage and recurrent pregnancy loss since 1983 in China (Permit No. Z44020008). To ensure clarity, we have replaced the current citation to the meta-analysis with the relevant guideline, including Guidelines for the Diagnosis and Treatment of Recurrent Pregnancy Loss with Integrated Traditional Chinese and Western Medicine (2023) and Clinical Guidelines for the Application of Chinese Patent Medicines in Treating Threatened Abortion (2024) (Chinese Journal of Integrated Traditional and Western Medicine. 2024;44(06): 645-659, DOI: 10.7661/j.cjim.20231225.122; Chinese Journal of Integrated Traditional and Western Medicine. 2024;44(11): 1285-1294, DOI:

10.7661/j.cjim.20241024.290).

19. Your comment 19: What are the side effects of ZYP? To what extent to the investigators believe the patients and personnel remained blind during the study?

Response: According to the instruction of ZYP, side effects of ZYP include nausea, dry mouth, and constipation that disappeared after drug withdrawal.

Regarding blinding, we took multiple measures to ensure that both participants and study personnel remained unaware of treatment allocation throughout the study. The placebo was identical to ZYP in appearance, packaging, and taste, minimizing the likelihood of unblinding due to sensory differences (please see **Supplement 3**). Randomization codes were securely stored and only accessible to an independent statistician not involved in patient care or assessment. Throughout the whole trial, blinding remained intact. Thus, blinding was well maintained for both patients and investigators.

Dear Reviewers,

Thank you very much for your constructive and insightful suggestions, which were extremely helpful in revising our manuscript. Below, we provide our point-by-point responses to your comments.

Response to Reviewer #1

1. Your comment 1a: 1. The objectives have now been added at the end of the introduction section, but this is not adequate. a. In the statement for the aim: ...Thus, the present study aimed to assess the efficacy...Since this is a pragmatic trial, please change “efficacy” to “effectiveness”

Response: We appreciate this valuable suggestion. We have revised the wording from “efficacy” to “effectiveness” throughout the manuscript to better reflect the pragmatic nature of our trial.

2. Your comment 1b: Please frame the specific objective using the PICO framework-see the CONSORT explanation document <https://doi.org/10.1136/bmj-2024-081124> for further guidance.

Response: In accordance with the 2024 CONSORT guideline, we have rephrased the study objective using the PICO structure. The revised objective now reads:

“Specifically, this multicenter, prospective, double-blind, placebo-controlled, randomized trial compared ZYP versus placebo in infertile women aged 35–42 years undergoing IVF/ICSI at 12 tertiary hospitals in China, to determine whether ZYP increased LBR in fresh ET cycles, and to evaluate its effects on ovarian stimulation (OS) outcomes, biochemical pregnancy rate, implantation rate, clinical pregnancy rate, miscarriage rate, and maternal, fetal, and neonatal safety.”

3. Your comment 2. Sample size: the effect size/detectable difference - You have explained in your response to my comment that your chosen effect size aligns with the general guidelines for the clinically important improvement, but this justification is not included in the main manuscript. Could you please add a sentence about this?

Response: We have added an explanation of how the chosen effect size of LBR aligned

with generally acknowledged clinically meaningful improvement in reproductive medicine before the sample size estimation was performed. We have added the following explanation to the Methods–Statistical Analysis section: “In the field of reproductive medicine, a plausible and clinically meaningful improvement of LBR is generally considered to be 5-10%. In our previous clinical research, the LBR was 0.42 per embryo transfer in ZYP-treated group and 0.33 per embryo transfer in placebo control group among AMA women, which aligned with this improvement and provided an empirically derived effect size of ZYP treatment. Thus, we performed sample size estimation based on this effect size....”

4. Your comment 3. Methods: You’ve explained the ITT set in the response document, but you haven’t clearly defined your full primary analysis set in the main manuscript. For example, on page 14 line 248: ...The primary analysis was performed according to the intention-to-treat (ITT) principle, i.e., analyzing all randomized participants...please clarify that the participants are analysed according to the group they were originally assigned.

Response: As suggested, we have revised the Methods–Statistical Analysis section to explicitly state that participants were analyzed according to the group to which they were originally assigned.

5. Your comment 4. Primary analysis: It is widely recognised in literature and published guidance (e.g. EMA/CHMP/295050/2013) that analytic methods should reflect the trial design, therefore adjustment should be made for the stratification variables in the primary analysis of trials that use stratified randomisation. This should be done by using a regression model suitable for your primary outcome. You’ve said you’ve done this as sensitivity analysis, but I think this should be the primary analysis.

Response: We appreciate the reviewer’s valuable comment and acknowledge that recent statistical guidance recommends incorporating stratification variables in the primary analysis model to reflect the trial design.

As part of the protocol, our pre-specified statistical analysis plan (SAP) defined a simple group comparison for fresh-cycle LBR as the primary analysis. Modifying this

post hoc would deviate from the approved protocol. However, to address your concern, we conducted an additional logistic regression including treatment group, site, and age. The adjusted model yielded consistent results with the unadjusted analysis (adjusted $P = 0.030$, OR = 1.33, 95% CI [1.03, 1.72]; see Supporting Information 2, Table S11).

We have described this in the Supporting Information and acknowledged in the Discussion that lack of covariate adjustment is a limitation.

6. Your comment 5. Statistical analysis: Handling of missing data: your chosen method of dealing with missing data i.e. based on the worst-case scenario, despite being conservative is prone to bias. You could explore the effect of missingness as sensitivity analysis using maximum likelihood-based methods such as multiple imputation.

Response: As recommended, we performed a sensitivity analysis using multiple imputation (five imputations) to assess the impact of missing live birth outcomes. Results were consistent with our primary analysis (see Supporting Information 2, Table S11).

7. Your comment 6. Choice of statistical power: We agree with the authors that both 80% and 90% power are acceptable, with 80% as the lowest benchmark, however, 90% is the common choice in any high quality RCT to reduce any chance of type II error. In fact, the Lancet paper you referenced (Lancet. 2019;393(10178):1310-1318. doi:10.1016/s0140-6736(18)32843-5) used 90% power as opposed to 80% as you state. It might be worth acknowledging that your choice of 80% power comes with a 20% risk of missing a true effect.

Response: We appreciate the reviewer's correction regarding the referenced Lancet trial and apologize for the oversight.

In our trial, we selected a power of 80%, which is generally considered the minimal acceptable benchmark for the power selection of clinical trials. We made this decision primarily for feasibility reasons, as achieving 90% power would have required a substantially larger sample size and extended recruitment period, which was not

practical within our multicenter study setting.

We selected 80% power for feasibility reasons, as 90% would have required a substantially larger sample size and extended recruitment. A systematic review (Hum Reprod, 2019, 34:659–665) found that only 10% of RCTs achieved 80% power to detect a 10% improvement in LBR, underscoring the challenge of powering trials for modest but meaningful effects. We have clarified this rationale in the Methods and acknowledged the trade-off in the Discussion.

Reviewers #2, 3 and 4

Response: We are pleased that our previous revision addressed all your concerns and that the updated manuscript now meets your expectations. We sincerely appreciate your review and constructive feedback.

Reviewer #5:

1. Your comment 1: In the response to the previous reviewers comments the authors state that “At first, in the current trial, adverse event monitoring revealed elevated ALT/AST in 2 participants from the ZYP group and 1 from the placebo group. One renal function related adverse event was reported in ZYP group.” But I cant see where this info is available in the manuscript or associated supp files? It should be there.

Response: We apologize for the confusion. Maternal adverse events during the treatment period are now reported in Supporting Information 2, Table S6.

2. Your comment 2: The authors state that some protocol changes were made during early days of recruitment, regarding the primary outcome and embryo quality definition. These should be transparently conveyed in the supplementary files/protocol.

Response: We agree with the importance of transparency in protocol amendments and have added a supplementary document titled “Protocol Revision History and Major Changes” (Supplement 10), detailing all substantial modifications, including version number, approval date, revised sections, specific changes, and rationale.

3. Your comment 3: Add as a footnote to table 3 that the live births (and presumably

all preg outcomes) exclude the 6 per-protocol pregnancies in each arm. Under pure ITT these would be included in the totals so people may be uncertain whether they have been or not.

Response: We appreciate your comment. As indicated in Figure 1, six pregnancies occurred after randomization but before ET in each arm (n = 12 total). These pregnancies were unrelated to the ET procedures and therefore excluded from the denominator when calculating fresh cycle live birth and other pregnancy-related outcomes. To enhance clarity, we have added a corresponding footnote to both Figure 1 and Table 3.

4. Your comment 4: Regarding blinding- if ZYP has specific side effects, patients can become unblinded despite the techniques used at your centre to conceal the allocations etc. It should be added to the limitations that the study was double-blind and placebo was used, but that it cant be ruled out there was some level of unblinding due to known side effects of ZYP.

Response: We appreciate the reviewer's concern regarding the potential for unblinding due to the known side effects of ZYP. According to the official drug instructions, ZYP may cause mild and transient side effects such as gastrointestinal discomfort (including nausea, vomiting, constipation, diarrhea, or stomach discomfort) and skin or subcutaneous reactions (such as rash or itching). Some patients may also experience dry mouth or a bitter taste.

However, it is important to note that these potential side effects overlap with symptoms commonly observed during ovarian stimulation procedures (e.g., gastrointestinal disturbances, abdominal distension, and skin reactions). Therefore, distinguishing treatment-related side effects from those associated with ovarian stimulation would be difficult for both patients and clinical personnel.

Additionally, adverse events were systematically recorded and are now summarized in Supplement 2, Table S6. The overall occurrence of adverse events was low in both groups. No clear pattern of treatment-related unblinding was observed. While we agree that unblinding due to the side effects cannot be fully excluded, the likelihood appears minimal. We have clarified this in the Discussion but do not consider it a limitation.

5. Your comment 5: I remain concerned that the per protocol analysis approach of excluding those not reaching ET is not the best approach. The authors justify it by a similar approach being used in two studies of fresh vs frozen transfer. This is a different scenario. In this ZYP study we anticipate possible effect of ZYP on the number/quality of embryos (this is probably not expected in the fresh vs frozen example). ZYP could have caused more cancellations in the control group. This is part of how the intervention might work, and not a case of ‘violating the protocol’ . But as long as the authors clearly explain the inclusions for the analysis, and seeing as it’s a secondary analysis anyway, I wont push it further.

Response: We appreciate the reviewer’s thoughtful comments and would like to further clarify the rationale behind our analysis set design.

We agree that treatment-related effects on cycle cancellation or embryo development may contribute to the overall intervention mechanism of ZYP. Therefore, in our trial, the primary analysis was performed in the ITT population, which included all randomized participants. As you also noted, this approach captured any potential treatment effects across the entire clinical course, including cycle cancellations and dropouts.

The PP analysis was designed to assess the direct treatment effect of ZYP on live birth among participants who completed embryo transfer, were willing to continue follow-up, and maintained good treatment compliance ($\geq 80\%$). Since the primary outcome was fresh-cycle live birth, our PP definition reflected participants who were both (1) able to achieve the outcome (i.e., underwent fresh ET) and (2) adhered to the assigned intervention per protocol. We believe this two-part definition aligns with the conventional interpretation of “per protocol,” including those who followed the study procedures sufficiently to evaluate the treatment’s efficacy under ideal adherence conditions.

We recognize that this approach may not fully capture the effects of ZYP on cycle progression. However, the PP analysis was intended as a complementary, rather than confirmatory, analysis to the ITT findings. To ensure transparency, we have revised Figure 1 in the previous version to clearly distinguish the ITT and PP populations.

Dear Reviewers,

Thank you so much for your constructive and insightful suggestions which were extremely helpful in our effort of revising our manuscript.

Response to Reviewers #1, 3 and 5

Response: We are very pleased that we previously addressed all your concerns and that our revised manuscript meets your expectations. We appreciate your expert review and constructive feedback.

Thanks to the authors for addressing my previous comments. Most of the comments have been addressed satisfactorily, but I've listed a few below which need to be readdressed.

1. The objectives have now been added at the end of the introduction section, but this is not adequate.
 - a. In the statement for the aim:Thus, the present study aimed to assess the **efficacy**.....

Since this is a pragmatic trial, please change "efficacy" to "effectiveness"
 - b. Please frame the specific objective using the PICO framework- see the CONSORT explanation document <https://doi.org/10.1136/bmj-2024-081124> for further guidance.
2. Sample size: the effect size/detectable difference - You have explained in your response to my comment that your chosen effect size aligns with the general guidelines for the clinically important improvement, but this justification is not included in the main manuscript. Could you please add a sentence about this?
3. You've explained the ITT set in the response document, but you haven't clearly defined your full primary analysis set in the main manuscript. For example, on page 14 line 248:The primary analysis was performed according to the intention-to-treat (ITT) principle, i.e., analyzing all randomized participants...please clarify that the participants are analysed according to the group they were originally assigned.
4. **Primary analysis:** It is widely recognised in literature and published guidance (e.g. EMA/CHMP/295050/2013) that analytic methods should reflect the trial design, therefore adjustment should be made for the stratification variables in the primary analysis of trials that use stratified randomisation. This should be done by using a regression model suitable for your primary outcome. You've said you've done this as sensitivity analysis, but I think this should be the primary analysis.
5. **Statistical analysis:** Handling of missing data: your chosen method of dealing with missing data i.e. based on the worst-case scenario, despite being conservative is prone to bias. You could explore the effect of missingness as sensitivity analysis using maximum likelihood-based methods such as multiple imputation.
6. **Choice of statistical power:** We agree with the authors that both 80% and 90% power are acceptable, with 80% as the lowest benchmark, however, 90% is the common choice in any high quality RCT to reduce any chance of type II error. In fact, the Lancet paper you referenced (Lancet. 2019;393(10178):1310-1318. doi:10.1016/s0140-6736(18)32843-5) used 90% power as opposed to 80% as you state. It might be worth acknowledging that your choice of 80% power comes with a 20% risk of missing a true effect.